# Mediterranean UNESCO World Heritage at risk from coastal flooding and erosion due to sea-level rise

Lena Reimann [1], Athanasios T. Vafeidis [1], Sally Brown [2,3], Jochen Hinkel [4] & Richard S.J. Tol [5]

UNESCO World Heritage sites (WHS) located in coastal areas are increasingly at risk from coastal hazards due to sea-level rise. In this study, we assess Mediterranean cultural WHS at risk from coastal flooding and erosion under four sea-level rise scenarios until 2100. Based on the analysis of spatially explicit WHS data, we develop an index-based approach that allows for ranking WHS at risk from both coastal hazards. Here we show that of 49 cultural WHS located in low-lying coastal areas of the Mediterranean, 37 are at risk from a 100-year flood and 42 from coastal erosion, already today. Until 2100, flood risk may increase by 50% and erosion risk by 13% across the region, with considerably higher increases at individual WHS. Our results provide a first-order assessment of where adaptation is most urgently needed and can support policymakers in steering local-scale research to devise suitable adaptation strategies for each WHS.

[1] Department of Geography, Kiel University, Ludewig-Meyn-Strasse 14, 24118 Kiel, Germany. [2] Faculty of Physical Sciences, University of Southampton, University Road, Highfield, Southampton SO17 1BJ, UK. [3] Department of Life and Environmental Sciences, Faculty of Science and Technology, Bournemouth University, Fern Barrow, Poole, Dorset BH12 5BB, UK. [4] Global Climate Forum e.V. (GCF), Neue Promenade 6, 10178 Berlin, Germany. [5] Department of Economics, University of Sussex, Falmer Campus, Brighton BN1 9SL, UK. Correspondence and requests for materials should be addressed to L.R. (email: reimann@geographie.uni-kiel.de)

Since 1972, the United Nations Educational, Scientific and Cultural Organisation (UNESCO) designates the world's common heritage under the World Heritage Convention[1]. The World Heritage List of 2018 comprises a total of 1092 cultural and natural heritage sites, based on their Outstanding Universal Value (OUV)[2]. Over 77% of these sites are cultural World Heritage sites (WHS) which have high intangible value as they represent icons of human civilisation[3,4]. A large share of cultural WHS are located in coastal areas as human activity has traditionally concentrated in these locations[5,6]. As the risk of coastal hazards such as flooding and erosion increases with sea-level rise (SLR)[7], a considerable number of coastal WHS will gradually be exposed to these hazards in the future[7,8], threatening the OUV of affected sites[9–12] and potentially leading to losses in economic revenue as WHS are popular tourist destinations[12,13]. This is particularly true for the Mediterranean region as several ancient civilisations have developed in the region[4,6,14], resulting in a high concentration of cultural WHS in coastal locations. Due to the small tidal range and steep topography in coastal areas, ancient and current settlements are often located directly at the waterfront and hardly above sea level[6,15]. Furthermore, adaptation methods and protection standards vary considerably across Mediterranean countries[16] due to large socioeconomic differences between northern, eastern and southern parts of the region[14,17], therefore leaving most WHS with limited protection from coastal hazards.

Although WHS are protected under the World Heritage Convention, countries themselves are responsible for their management, which includes adaptation to climate change[18]. However, WHS management plans rarely consider adaptation to SLR impacts[11,19]. Although climate change has been acknowledged as a threat to WHS in recent years[3,9,19,20], few studies have explored this aspect, leaving heritage managers and policymakers with little information on potential adaptation options. Therefore, previous work has called for more research identifying WHS at risk to inform adaptation planning and to ensure that their OUV is preserved[9–11,18,20,21]. It has expressed the need for more robust data and modelling approaches on local to regional scales, as adaptation planning takes place at a national level and specific adaptation measures are implemented at a local level[9,11,22]. The results of assessments based on these methods can support adaptation planning, especially in prioritising adaptation strategies with limited financial resources[3,8,12,19,22,23].

Previous studies have primarily focused on local-scale assessments of various climate change impacts on UNESCO WHS[11,12,19,22,24–26] or on natural hazards, such as landslides and river floods, without directly considering climate change[13,27–30]. To our knowledge, only one large-scale study has analysed the long-term impacts of SLR on cultural UNESCO WHS[7]. This study was based on aggregate WHS data provided on the UNESCO website, where every WHS is depicted by a point that represents its approximate centre, even if the WHS consists of a number of so-called serial nominations[31]. Consequently, the location of the point can substantially deviate from the location of the actual WHS. Further, none of the above-mentioned studies assessed the risks of coastal flooding due to extreme sea levels (ESL) or to coastal erosion due to SLR.

To address the current research gap, we assessed Mediterranean UNESCO cultural WHS at risk from coastal flooding and erosion under four SLR scenarios from 2000 to 2100. We used an index-based approach that allows for ranking and comparing WHS at risk. For this purpose, we produced a WHS dataset containing spatially explicit representations of all Mediterranean WHS located in low-lying coastal areas. Results show that the vast majority of WHS at risk from either of the two hazards until 2100 are already at risk under current conditions. Risk will increase in the course of the century, its magnitude depending on the rate of SLR, with particularly high increases in coastal flood risk and at individual WHS. Our results can support adaptation planning in determining potential risk thresholds (tipping points) based on the temporal evolution of the indices. Additionally, based on the WHS most at risk policymakers can designate priority areas for further analysis in order to devise specific adaptation strategies.

## Results

**UNESCO World Heritage in coastal areas**. The modified and extended WHS dataset[32] comprises 159 data entries that represent inscribed (main) WHS (49) along with their serial nominations (110) located in the Mediterranean Low Elevation Coastal Zone (LECZ), which is defined as all land with an elevation of up to 10 m in hydrological connection to the sea[33]. The data comprise attributes adopted from the original dataset and newly added attributes (e.g. heritage type, elevation, WHS location in urban settlements, distance from the coast). See Supplementary Table 1 for a complete list of attributes. Our analysis focuses on an aggregated version of the dataset that contains the 49 main WHS. Figure 1 shows the 49 main WHS located in the Mediterranean LECZ. Approximately one third of these WHS are located in Italy (15), followed by Croatia (7), Greece (4), and Tunisia (4). In most instances, only certain parts of the WHS (on average 35%) fall into the LECZ; five sites are fully located in the LECZ (see dataset).

**Flood risk**. Under current conditions (base year 2000), 37 WHS are at risk from ESL, defined as the 100-year storm surge (including tides) plus the amount of SLR for the respective scenario and year (see Methods), which corresponds to 75% of all sites located in the LECZ. This number increases to 40 WHS at risk under the high-end (HE) scenario. The flood area ranges from 0.03% of the total WHS at Archaeological Site of Leptis Magna (183) and Cultural Landscape of the Serra de Tramuntana (1371) to 97% at Venice and its Lagoon (394), with a mean of 11.3%. The average flood area increases to over 14% in 2100 under the HE scenario, corresponding to an increase of 24% compared to 2000. Under Representative Concentration Pathway 2.6 (RCP2.6), RCP4.5 and RCP8.5, the average flood area increases to around 12% in 2100 (Fig. 2a). In 2000, the highest flood depth of 1.2 m can be found at Archaeological Area and the Patriarchal Basilica of Aquileia (825) while the mean of maximum flood depth for all sites amounts to roughly 0.4 m. The maximum flood depth increases by approximately 70% to a mean of more than 0.6 m under RCP2.6, 92% (over 0.7 m) under RCP4.5, 121% (approximately 0.8 m) under RCP8.5 and 290% (roughly 1.5 m) under the HE scenario (Fig. 2b), where the highest flood depth of 2.5 m can be found at Venice and its Lagoon (394). The flood risk index that results from combining flood area and flood depth (see Methods) has a mean of 3.7 in 2000, which increases by 25% to 4.6 under RCP2.6 and by almost 50% to 5.5 under the HE scenario (Fig. 2c).

In the base year, the risk index ranges from 0 for those sites that are not at risk to a maximum of 10 at Venice and its Lagoon (394), Ferrara, City of the Renaissance, and its Po Delta (733) and Archaeological Area and the Patriarchal Basilica of Aquileia (825). These WHS are located along the northern Adriatic Sea where ESL are highest as high storm surges coincide with high regional SLR (Fig. 1 and Supplementary Figure 1). Under the HE scenario, a total of six WHS have the highest risk index of 10, four of which are located in Italy and two in Croatia (Fig. 3). In 16 Mediterranean countries (including Gibraltar), at least one WHS is at risk under at least one of the four scenarios. The highest number of WHS at risk can be found in Italy (13), which

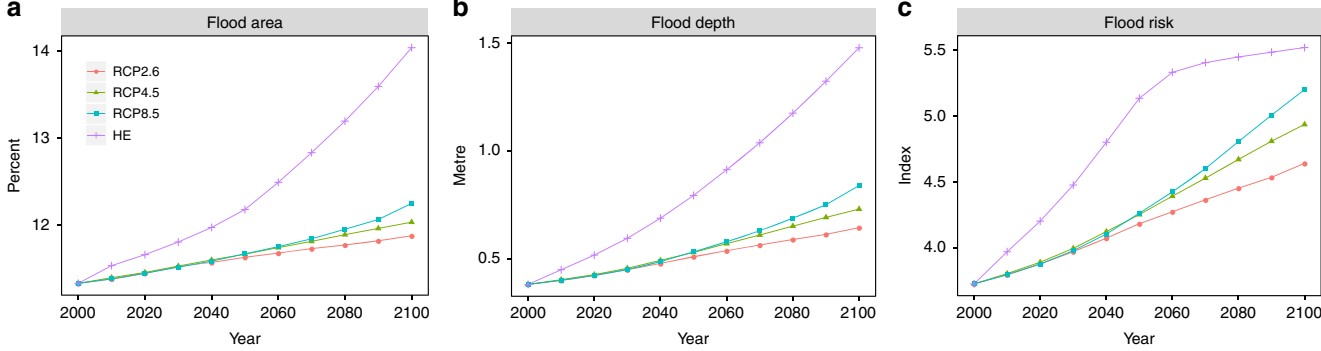

**Fig. 1** UNESCO cultural World Heritage sites located in the Mediterranean Low Elevation Coastal Zone (LECZ). All sites are shown with their official UNESCO ID and name. The map also shows extreme sea levels per coastal segment based on the Mediterranean Coastal Database[108] under the high-end sea-level rise scenario in 2100

| ID | Name |
|---|---|
| 36 | Medina of Tunis |
| 37 | Archaeological Site of Carthage |
| 79 | Paphos |
| 95 | Old City of Dubrovnik |
| 97 | Historical Complex of Split with the Palace of Diocletian |
| 125 | Natural and Culturo-Historical Region of Kotor |
| 131 | City of Valletta |
| 164 | Arles, Roman and Romanesque Monuments |
| 183 | Archaeological Site of Leptis Magna |
| 184 | Archaeological Site of Sabratha |
| 193 | Tipasa |
| 295 | Byblos |
| 299 | Tyre |
| 332 | Punic Town of Kerkuane and its Necropolis |
| 356 | Historic Areas of Istanbul |
| 394 | Venice and its Lagoon |
| 395 | Piazza del Duomo, Pisa |
| 484 | Xanthos-Letoon |
| 493 | Medieval City of Rhodes |
| 498 | Medina of Sousse |
| 530 | Delos |
| 565 | Kasbah of Algiers |
| 570 | Butrint |
| 595 | Pythagoreion and Heraion of Samos |
| 712 | City of Vicenza and the Palladian Villas of the Veneto |
| 726 | Historic Centre of Naples |
| 733 | Ferrara, City of the Renaissance, and its Po Delta |
| 788 | Early Christian Monuments of Ravenna |
| 809 | Episcopal Complex of the Euphrasian Basilica in the Historic Centre of Poreč |
| 810 | Historic City of Trogir |
| 825 | Archaeological Area and the Patriarchal Basilica of Aquileia |
| 826 | Portovenere, Cinque Terre, and the Islands (Palmaria, Tino and Tinetto) |
| 829 | Archaeological Areas of Pompei, Herculaneum and Torre Annunziata |
| 830 | Costiera Amalfitana |
| 842 | Cilento and Vallo di Diano National Park with the Archeological Sites of Paestum and Velia, and the Certosa di Padula |
| 875 | Archaeological Ensemble of Tárraco |
| 963 | The Cathedral of St James in Šibenik |
| 978 | Old Town of Corfu |
| 1018 | Ephesus |
| 1024 | Late Baroque Towns of the Val di Noto (South -Eastern Sicily) |
| 1042 | Old City of Acre |
| 1096 | White City of Tel-Aviv --the Modern Movement |
| 1200 | Syracuse and the Rocky Necropolis of Pantalica |
| 1211 | Genoa: Le Strade Nuove and the system of the Palazzi dei Rolli |
| 1220 | Bahá'i Holy Places in Haifa and the Western Galilee |
| 1240 | Stari Grad Plain |
| 1371 | Cultural Landscape of the Serra de Tramuntana |
| 1500 | Gorham's Cave Complex |
| 1533 | Venetian Works of Defence between 15th and 17th centuries: Stato da Terra – western Stato da Mar |

**Fig. 2** Temporal evolution of the flood risk indicators at each World Heritage site, averaged across the Mediterranean region. Results are shown from 2000 to 2100 for RCP2.6, RCP4.5, RCP8.5 and the high-end (HE) scenario. **a** Mean area flooded (in %), **b** mean flood depth (in m) and **c** mean flood risk index

corresponds to 87% of the Italian WHS located in the LECZ, followed by Croatia (6; 86%) and Greece (3; 75%). See also Supplementary Figure 2 for the flood risk indicators at each WHS and Supplementary Data 1 for the raw data of the indicators.

**Erosion risk**. Under current conditions, 42 WHS are at risk from coastal erosion, which corresponds to 86% of all sites located in the LECZ. This number increases to 46 WHS under the HE

scenario. Erosion risk is predominantly determined by the distance of a WHS from the coastline. Already in the base year, 31 WHS are at least partly located within 10 m of the coastline, which increases to 39 sites under the HE scenario (Supplementary Figure 4), based on the assumption that all areas below the amount of SLR are permanently inundated (see Methods). The average distance from the coast decreases from roughly 1.1 km in 2000 by 30% to 762 m under RCP2.6 and by more than 90% to slightly above 100 m under the HE scenario (Fig. 4a). As we

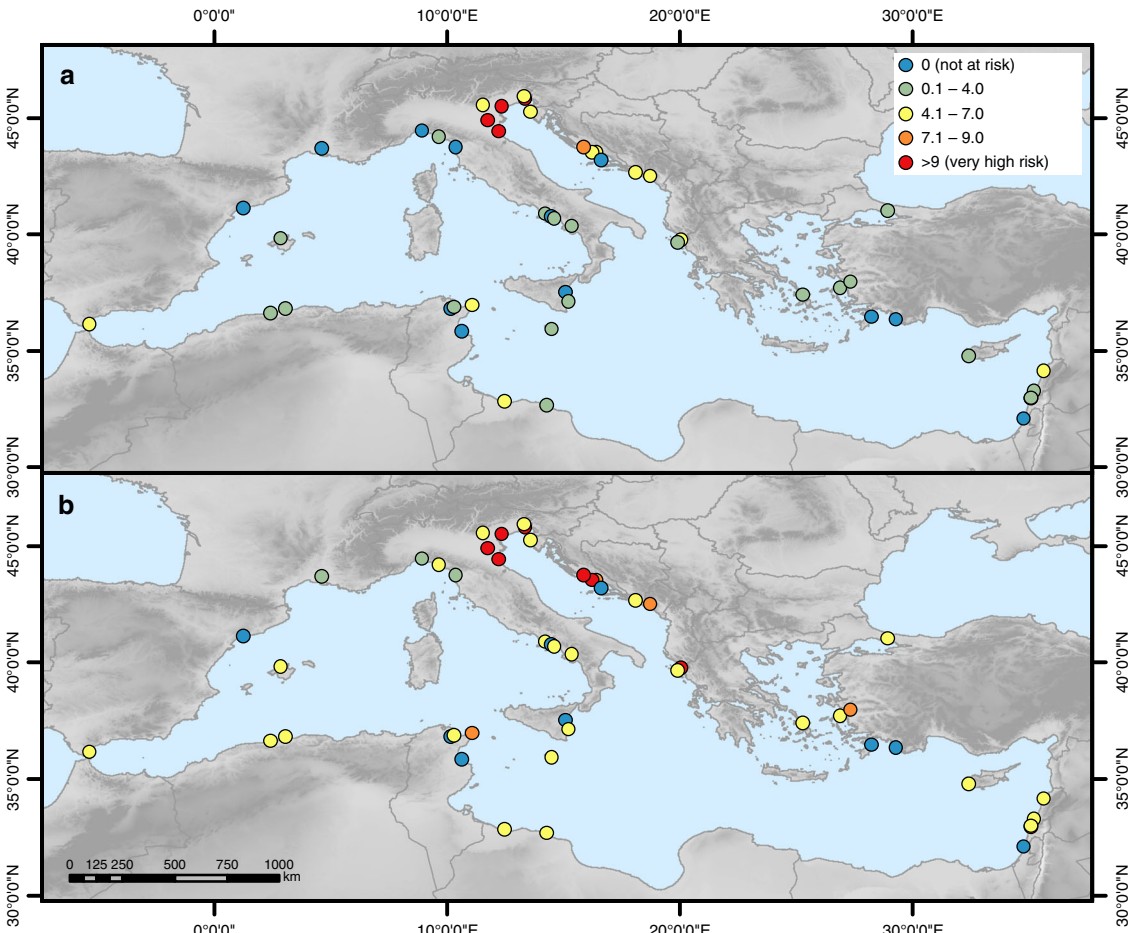

**Fig. 3** Flood risk index at each World Heritage site under current and future conditions. **a** In 2000 and **b** in 2100 under the high-end sea-level rise scenario

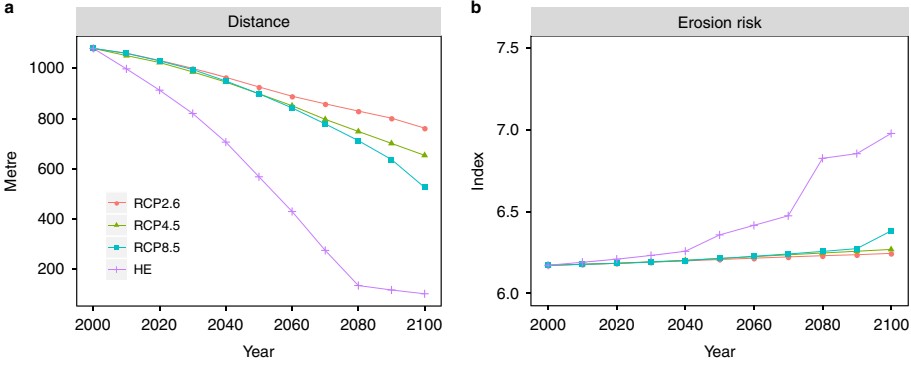

**Fig. 4** Temporal evolution of the dynamic erosion risk indicators at each World Heritage site, averaged across the Mediterranean region. Results are shown from 2000 to 2100 for RCP2.6, RCP4.5, RCP8.5 and the high-end (HE) scenario. **a** Mean distance from the coastline (in m) and **b** mean erosion risk index

assume the erosion risk indicators coastal material, mean wave height and sediment supply to remain constant in the course of the century, the erosion risk index increases only slightly from 2000 to 2100. The average erosion risk index increases from 6.2 in 2000 to 6.3 in 2100 under RCP2.6 and RCP4.5. Under RCP8.5 it increases to 6.4 and under the HE scenario it increases to 7, which corresponds to an increase of 13% compared to 2000 (Fig. 4b).

In the base year, the erosion risk index ranges from 0 for those sites not at risk to 9.8 (very high) at Tyre (299) (Fig. 5), which is located directly at the coastline (very high risk) and is characterised by sandy material (very high risk), a mean wave height of 0.7 m (high risk) and sediment supply of just below 1 mg l$^{-1}$ (high risk).

The second highest risk index can be found at Pythagoreion and Heraion of Samos (595). Under the HE scenario, erosion risk remains highest at Tyre, followed by Archaeological Ensemble of Tárraco (875), Pythagoreion and Heraion of Samos (595) and Ephesus (1018), all of which have a very high index of 9 and higher. Similar to flood risk, in 16 Mediterranean countries (including Gibraltar) at least one WHS is at risk from coastal erosion under at least one of the four scenarios. The highest number of WHS at risk can be found in Italy (14), which corresponds to 93% of the Italian WHS located in the LECZ, followed by Croatia (7; 100%) and Greece (4; 100%). Erosion risk varies moderately across the Mediterranean region and no regional pattern can be discerned as

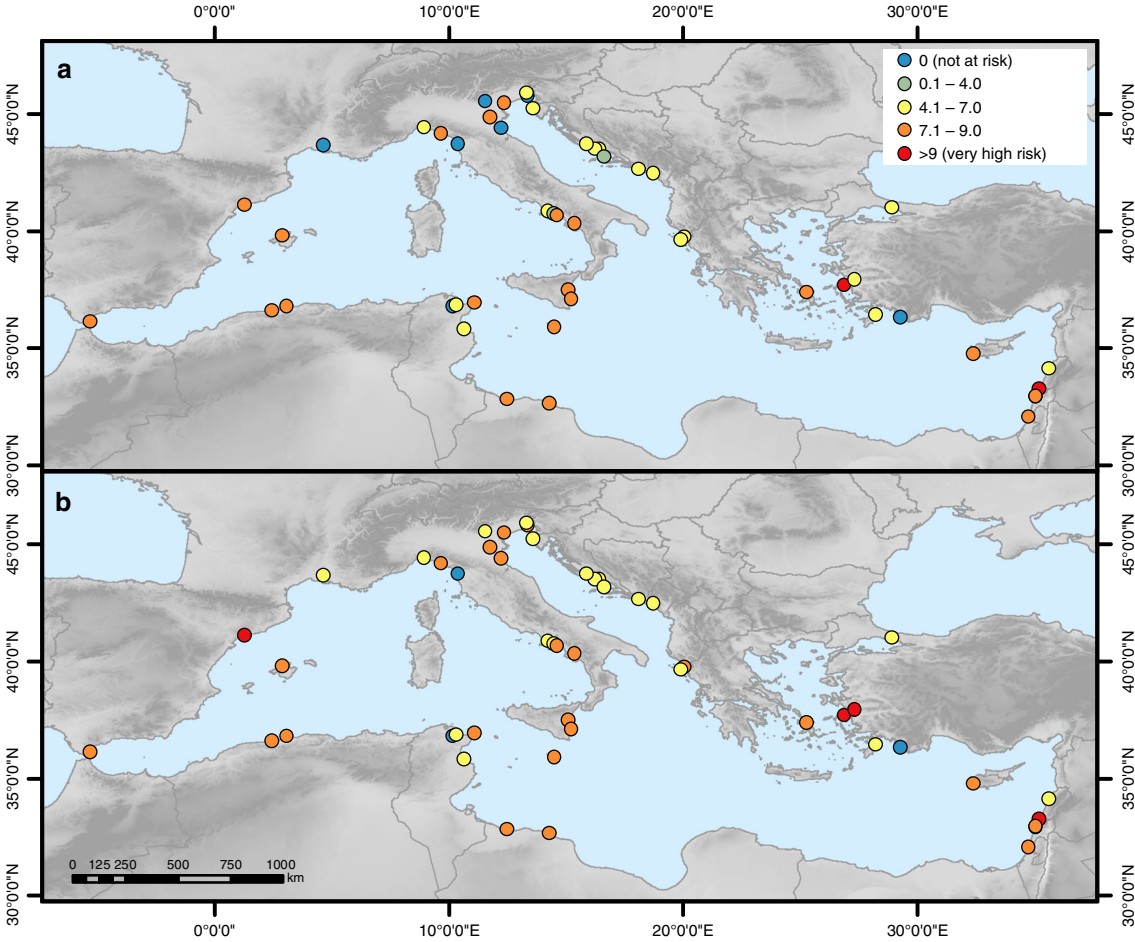

**Fig. 5** Erosion risk index at each World Heritage site under current and future conditions. **a** In 2000 and **b** in 2100 under the high-end sea-level rise scenario

erosion risk indicators are mostly site-specific. (Please see Supplementary Figure 3 and Supplementary Figure 4 for the erosion risk indicators at each WHS and Supplementary Data 2 for the raw data of the indicators.)

## Discussion

In this study, we assess UNESCO WHS at risk from coastal flooding and erosion under four SLR scenarios until 2100, based on revised and extended spatially explicit WHS data. The use of an index-based approach enables a quick evaluation of both risks that can easily be applied to other locations[34–36]. With the help of the risk indices, we are able to rank and compare WHS, while at the same time we avoid attaching a monetary value to them[37]. The results of this study can therefore support adaptation planning at different spatial scales: at the national scale, especially in countries with a large number of WHS at risk such as Croatia, Greece, Italy and Tunisia; at the EU scale, as, for example, regulated under the EU Floods Directive[38]; and at the basin scale, as prescribed under the Barcelona Convention, which is the basis for the Mediterranean Action Plan and the Protocol on Integrated Coastal Zone Management (ICZM) in the Mediterranean[39]. Our results can be particularly useful in designating priority areas with urgent need for adaptation and can serve as a basis for further, more in-depth assessments[40]. Furthermore, the temporal evolution of the risk indices and their individual components can provide valuable information on the point in time when a WHS may be at risk or when a certain risk threshold may be exceeded[23]. This threshold can be referred to as an adaptation

tipping point as its exceedance requires a (new) policy action[41,42]. An example of such potential tipping points for both risk indices is shown in Fig. 6. These insights can be used to ensure that the OUV of WHS at risk from either of the two hazards is preserved in the long term.

In total, 47 WHS may be at risk from at least one of the two hazards by the end of the century, with Piazza del Duomo, Pisa (395) potentially at risk from flooding only and seven sites (UNESCO IDs 493, 498, 829, 975, 1024, 1096, 1240) from erosion only. Based on these results, only two sites, Medina of Tunis (36) and Xanthos-Letoon (484), are not at risk from any of the two hazards by 2100. Further, we find that 93% of the sites at risk from a 100-year flood and 91% of the sites at risk from coastal erosion under any of the four scenarios are already at risk under current conditions, which stresses the urgency of adaptation in these locations.

Risk will further increase by 2100, in particular in the second half of the century, when projections of SLR diverge considerably based on the respective scenario. Therefore, the magnitude of risk increase largely depends on global mitigation efforts in the next years, which should pursue the aim not to exceed RCP2.6[43] as planned under the Paris Agreement[44] (projections based on RCP2.6 are closest to the 1.5 °C goal of the Paris Agreement). If the goal of the Paris Agreement is not met, the amount of SLR may exceed the height of a 100-year storm surge by a factor of 1.4 under RCP8.5 and a factor of 3 under the HE scenario in 2100. Therefore, SLR may become a larger threat to WHS than a present-day 100-year storm surge. A recent study of future ESL at

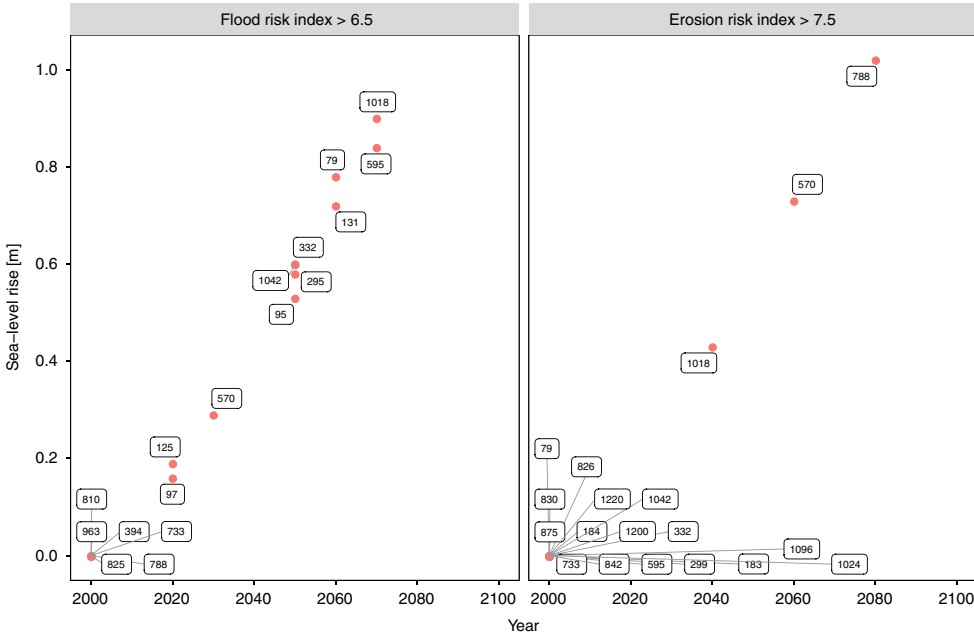

**Fig. 6** Examples of potential adaptation tipping points for the flood risk index and the erosion risk index. Both graphs show points in time when a World Heritage site may exceed a certain risk threshold with the respective amount of sea-level rise under the high-end scenario. Point labels show the official UNESCO ID of the sites affected. **a** Flood risk index threshold of 6.5 and **b** erosion risk index threshold of 7.5

the European scale has come to similar results, suggesting that present-day 100-year events in the Mediterranean may occur much more frequently, up to several times per year, by 2100[45]. Our results illustrate the value of rigorous global-scale mitigation efforts which could be crucial in preventing WHS from losing their OUV, especially as protection measures only work effectively up to a certain water level. Recent research has shown that RCP2.6 may be exceeded by 2100[46–48], therefore adaptation planning should prepare for higher SLR scenarios.

As adaptation measures need to be integrated into the WHS without compromising its OUV, adaptation planning at WHS is particularly challenging[11,49]. Since a site's OUV is bound to its location, retreat seems to be the least favourable adaptation option[11,19,24]. While relocation of individual monuments such as the Early Christian Monuments of Ravenna (788) or The Cathedral of St. James in Šibenik (963) may be technically possible, it seems to be impossible to relocate WHS that extend over large areas such as urban centres, archaeological sites and cultural landscapes. Examples of non-UNESCO cultural heritage monuments that have been moved inland are Clavell Tower[50] and Belle Tout lighthouse[51] in the UK and Cape Hatteras Lighthouse in the USA[52]. However, we could not find any examples in the existing literature where a UNESCO WHS was relocated. Relocation should be assessed carefully on a case-by-case basis and may be a suitable adaptation strategy for those WHS where risk is very high.

Common accommodation strategies such as hazard insurance, emergency planning or land-use planning[53] cannot be applied to WHS, but strategies to raise awareness can be pursued. Terrill[3] suggests to use the iconic nature of WHS to emphasise the severity of their loss in order to raise awareness of policymakers and heritage managers and to promote climate change mitigation[3]. Recent efforts at the national to local level that monitor cultural heritage and provide guidance for managing heritage in the light of climate change show that awareness is gradually increasing. Examples are the Irish Heritage Council, Historic England, the US National Park Service's Cultural Resources Climate Change Strategy and the Scottish Coastal Heritage at Risk project, which has developed a smartphone app for surveying

cultural heritage at risk from coastal erosion. This project raises awareness of local communities and authorities who can help designate priority areas and can therefore support heritage management[54]. Further, Khakzad et al.[55] suggest to include coastal heritage into ICZM, which may help in increasing the efficiency of adaptation planning. Another accommodation strategy would be to remove the inventory of WHS, such as paintings or statues, during flood events.

Coastal protection seems to be a suitable adaptation strategy as it may be possible to integrate it into any type of cultural WHS (i.e. urban heritage, archaeological site, cultural landscape or monument) without compromising its OUV. One example is the MOSE (Modulo Sperimentale Elettromeccanico/Experimental Electromechanical Module) project currently under construction in Venice (www.mosevenezia.eu). The entire lagoon will be protected by submerged mobile barriers at the lagoon inlets that will be raised during high waters of at least 1.1 m. These barriers do not interfere with the appearance of Venice and the fragile ecosystem of the lagoon as long as they are not raised frequently[18,49]. This example illustrates that, in order to preserve the aesthetic value of a WHS, very expensive protection measures may have to be pursued. An alternative to hard protection measures may be the use of coastal ecosystems as soft, nature-based protection by attenuating water levels and stimulating sedimentation in certain locations[56,57].

A combination of awareness-raising strategies and protection measures seem to be the most suitable adaptation strategies, but relocation also needs to be considered, in particular where risk is very high. However, local-scale assessments are needed in order to devise adaptation measures that are tailored to the characteristics of individual WHS and the type of hazard they are at risk from[11,19]. With regard to flood risk, such local-scale assessments should additionally consider a potential low bias in return flood heights due to uncertainties regarding the rate of SLR to avoid an underestimation of risk in the adaptation process[58].

As a first-order risk assessment, using a simple methodology based on publicly available region-wide data, this study can easily be reproduced and applied to other regions where a high number

of WHS is potentially at risk from coastal hazards due to SLR (e.g. South-East Asia). However, such assessments should bear in mind the limitations of this study. We have refrained from analysing the vulnerability of WHS to the two hazards as local-scale data concerning the internal characteristics of a WHS such as heritage material or heritage inventory are not readily available and including those in the analysis goes beyond the scope of this first-order assessment. Furthermore, we regard the use of depth-damage functions that are commonly applied in large-scale flood risk assessments to represent vulnerability[59–64] as problematic in the context of UNESCO World Heritage. Due to the high intangible value of WHS[3,11], it is very difficult and ethically questionable to quantify the damages at a WHS, which would imply that one WHS is more valuable than another[12]. However, if appropriate local-scale data are available, it may be possible to assess the tangible costs of coastal flooding and erosion by accounting for, for example, loss of revenue or cost of repairs[65].

The elevation-based (bathtub) approach used for modelling the floodplain tends to overestimate the flood extent, in particular in low-lying, mildly sloping terrain such as the Nile, Rhone and Po deltas[66,67], as hydrodynamic and hydraulic processes are not considered[36,68,69]. However, in steep terrain the flood extent is only slightly overestimated or even underestimated[66–68]. As large parts of the Mediterranean are characterised by steep topography[6], we expect this approach to provide a reasonable approximation of maximum potential flood extent at the majority of WHS. Furthermore, this modelling approach is extensively used in large-scale flood modelling[60–62,70–73] and can be regarded as a standard in such assessments[35,74].

As we do not consider defence structures in place due to lack of data on coastal protection measures[16], we may additionally overestimate risk in locations where protection measures exist. This appears to be the case at the Early Christian Monuments of Ravenna (788) and Archaeological Area and the Patriarchal Basilica of Aquileia (825), both located along the northern Adriatic Sea, where flood risk is modelled to be very high and erosion risk is modelled to increase rapidly at the end of the century, even though these WHS are currently located 6.7 and 3.5 km inland (Supplementary Data 2). A further example is Venice and its Lagoon (394), which is, according to our results, one of the WHS most at risk from coastal flooding (Fig. 3) and erosion (Fig. 5) until 2100. However, once construction of the MOSE project is completed (expected in 2018 as of the last official status[75]), risk will be reduced considerably as the flood barriers will protect the city and the lagoon from ESL of up to 3 m (www.mosevenezia.eu). According to our results, this protection level will be sufficient until 2100, with ESL projected to be 2.5 m under the HE scenario. As Venice has struggled with flood waters for centuries[49], it forms a special case; we did not find any other Mediterranean example where protection measures have been installed to protect an entire WHS.

We must also note that we may underestimate the floodplain in certain locations as it was not possible to account for human-induced subsidence even though it can be high in cities[76,77] such as Venice[78] and Istanbul[79] and in river deltas such as those of the Nile, Po and Rhone[80,81] due to ground water extraction. Currently, there is a lack of consistent data and of reliable scenarios projecting future development of human-induced subsidence[60]. Furthermore, the Shuttle Radar Topography Mission (SRTM) digital elevation model (DEM) used is a surface model and as such it may overestimate elevation in forested and built-up areas[82,83]. We observe this effect in Venice and its Lagoon (394) where only small sections of the city's built-up areas are located at elevation increments of 1–3 m AMSL, although the City of Venice reports the island to be almost fully inundated (91%) during a flood of 2 m[84]. A second example is Ferrara, City of the

Renaissance, and its Po Delta (733) where forest directly located at the coast[70] has elevation values of more than 10 m. Across the whole Mediterranean, built-up areas make up over 75% of the WHS located in the LECZ (see dataset), potentially leading to an underestimation of elevation, and therefore the risks of flooding and erosion in these locations. Despite its limitations, the SRTM DEM is currently the most consistent and commonly used global elevation model[85] and we did not have access to any other higher-resolution region-wide DEM as LiDAR (Light Detection And Ranging) data are only available for certain parts of the Mediterranean and the newly created CoastalDEM[86] is not freely available. Please consult Kulp and Strauss[85] for an in-depth discussion of the SRTM limitations.

The limitations of this study can be addressed in local-scale assessments that should be conducted to develop specific adaptation strategies and to select suitable adaptation measures for individual WHS. We encourage other researchers to use the revised and extended WHS data as a starting point for such assessments that allow for applying hydrodynamic modelling approaches, including higher-resolution local-scale data, and accounting for vulnerability.

Our results can raise awareness of policymakers and heritage managers by pointing to the urgent need for adaptation as a large number of WHS are already at risk from coastal flooding and erosion under current conditions. Both risks will exacerbate in the course of the twenty-first century and possibly beyond, their magnitude depending on the global-scale mitigation effort in the coming years. However, adaptation can only be implemented to a limited degree, especially with regard to WHS, as their OUV may be compromised by adaptation measures. If no steps are taken, WHS may lose their OUV in the next centuries and may consequently be removed from the UNESCO World Heritage list. Therefore, mitigation efforts are as much needed as adaptation to protect our common heritage from being lost. As UNESCO WHS are monitored at least to a certain degree under the World Heritage Convention, they will more likely receive the necessary attention and funding for adaptation measures against the risks of SLR. This is particularly true for WHS in densely populated locations such as the cities of Venice, Dubrovnik, Tyre or Tel-Aviv due to the high potential impacts of coastal hazards[23,60]. Cultural heritage not inscribed in the World Heritage list will receive much less attention and many of these heritage sites will slowly disappear with SLR even though these sites are important parts of human history as well[23].

## Methods

**General framework**. We employ the conceptual risk framework of the Intergovernmental Panel on Climate Change (IPCC) widely used in the current literature[61,62,87–89], in which risk results from the interaction of hazard, exposure and vulnerability[90,91]. To assess coastal flood risk, we define hazard as the intensity (i.e. surge height) and frequency (i.e. return period) of a storm surge and exposure as the area of a WHS flooded, along with the flood depth. To assess the risk of coastal erosion, we define the amount of SLR as the hazard and determine exposure of a WHS to coastal erosion by the distance of a WHS from the coast, combined with the characteristics of the coastal zone that determine its sensitivity to coastal erosion. We do not assess a site's vulnerability to either coastal flooding or erosion as analysis of the internal characteristics of a WHS, such as heritage material and inventory, are needed. Such data are not readily available, and therefore this work is beyond the scope of this regional assessment.

In order to quantify flood risk and erosion risk we use an index-based approach, which is a well-established method in the literature[34,92–99] and particularly suitable for first-order assessments on regional scale to support adaptation planning[40,93,99]. With the help of the risk indices we are able to assess potential impacts on WHS with rising sea levels and compare WHS with each other without attaching monetary value to them[37]. For transparency reasons and to ease application of our methodology to other regions, we select risk indicators that are based on publicly available data. An overview of the data used can be found in Supplementary Table 2.

**UNESCO World Heritage data processing**. We use the UNESCO World Heritage List data of 2018 provided on the UNESCO website[2], in which each WHS is

represented as a point, with longitude and latitude coordinates. We extract all cultural WHS located along the Mediterranean Sea. To account for WHS consisting of more than one site, so-called serial nominations[31], we manually check each WHS and add further point data entries for serial sites based on maps and descriptions provided on the UNESCO website[2]. To reflect each WHS location as accurately as possible, we follow the methodology used in Chang et al.[100] and Dassanayake et al.[101]. Therefore, we correct the location of misplaced WHS by using Google Earth™ satellite imagery. Where in doubt, we additionally compare photos and site descriptions provided on the UNESCO website with photos of the Panoramio web service embedded in Google Earth™ (as of January 2018 replaced by photos from Google Maps). Next, we examine WHS maps downloaded from the UNESCO website and digitise the outline of each site with the help of Google Earth™, resulting in one polygon for each serial WHS. We validate our WHS polygons by comparing them to those produced as part of the European PRO-THEGO project, available in a map viewer[102].

Subsequently, we extract the WHS located in the LECZ based on the lowest elevation value of each WHS polygon in the SRTM DEM version 4.1[103,104]. The LECZ represents all land with an elevation of up to 10 m in hydrological connection to the sea[33]. This way we ensure that all sites potentially exposed to coastal flooding and erosion are included in the analysis.

**Flood risk**. To assess WHS at risk from ESL, we calculate the floodplain of a storm surge with a 100-year return period under four SLR scenarios from 2000 to 2100. We use a 100-year storm surge as it is a standard measure for coastal protection and has been widely used in previous assessments[60–62,72,73,76,77,105–107]. To account for spatial differences in the floodplain across the Mediterranean basin, we use storm surge data from the Mediterranean Coastal Database (MCD)[108,109], where surge heights are available for each of the approximately 12,000 coastal segments. We select surge heights that are derived from the Global Tide and Surge Reanalysis (GTSR) dataset which accounts for ESL due to storm surges and tides. A detailed description of the methods used for developing the dataset can be found in Muis et al.[72]. In the MCD, a downscaled version of the GTSR data is available. To ensure that all data used for the analysis are referenced to the same vertical datum, we convert the vertical datum of the surge data, referenced to the mean sea level, to the EGM96 geoid, the vertical datum of the SRTM data[68,73,85,86]. To do so, we use the mean dynamic ocean topography[110], which is the difference between mean sea level and the geoid.

To account for plausible increases in ESL due to SLR, we combine the adjusted surge heights with four SLR scenarios based on the Representative Concentration Pathways (RCPs)[111]. We use the regionalised SLR projections by Kopp et al.[112] that account for three ice-sheet components, glacier and ice cap surface mass balance, thermal expansion and other oceanographic processes, land water storage and non-climatic factors such as Glacial Isostatic Adjustment[112,113]. These projections are available as grid points with a spatial resolution of 2° by 2°. We select the median projections (50th percentile) of RCP2.6, RCP4.5 and RCP8.5 for 2010–2100 to cover the likely range of uncertainty regarding SLR, as well as the 95th percentile of RCP8.5 (5% probability) to account for a HE scenario. We spatially join the grid points of the SLR projections to the coastal segments of the MCD closest to each

point and calculate the ESL of a 100-year storm surge for each coastal segment, scenario and 10-year time step. We do not account for potential changes in storminess as confidence in these projections is low[114].

We model the 100-year coastal floodplain for each SLR scenario with the help of a planar elevation-based (bathtub) approach using the SRTM DEM, which is extensively used in large-scale flood modelling[60–62,70,72,73]. The SRTM data used have a spatial resolution of 3 arc seconds (approximately 90 m at the equator) and a vertical resolution of 1 m[104]. Based on these data, we determine the area of each WHS located at elevation increments from 0 m up to 4 m in hydrological connection to the sea in a first step. Next, we attribute the calculated ESL to the nearest WHS. If more than one ESL can be attributed to one WHS, we calculate a weighted mean based on the number of raster cells with a specific ESL height assigned to each WHS. To determine the area of each WHS flooded (in %), we linearly interpolate between respective elevation increments based on the ESL assigned, following the method of Hinkel et al.[60]. We further calculate the maximum flood depth per WHS (in m) based on the difference between the ESL and the elevation value in the SRTM DEM. For WHS located below 0 m according to the SRTM data, we assume the minimum elevation value of each WHS to be 0 m. We apply this assumption to correct for artefacts present in the SRTM data, such as individual pixels with very low-elevation values (e.g. −20 m at Venice and its Lagoon (394))[115]. Using these values would result in unrealistically high maximum flood depths. Further, we do not account for existing flood protection measures in our analysis due to a lack of consistent region-wide data. Data of existing flood defences may be available for specific locations across the region, but integrating those into our analysis would compromise the consistency of our results.

For the flood risk index, we scale flood area and flood depth linearly to values ranging from 0 (not at risk) to a maximum value of 5 (very high risk), assuming that a WHS is at very high risk when at least 50% of the site are flooded with a flood depth of at least 1 m[60,116] (Table 1). We must note that we could not find any studies assessing flood risk based on the area of an object flooded; therefore, we assume that the OUV of a WHS is seriously threatened if at least half of the site is flooded. In a last step, we calculate the sum of the scaled flood risk indicators, which results in an index ranging from 0 to 10.

**Erosion risk**. To analyse WHS at risk from coastal erosion due to SLR, we calculate an erosion risk index for each WHS from 2000 to 2100 under the four SLR scenarios (RCP2.6, RCP4.5, RCP8.5, HE). We adopt the indicators used in previous index-based approaches on coastal erosion[40,92–94,96,117,118] and cultural heritage at risk from coastal erosion[5,34,95] and select those that play a key role in the Mediterranean[119] and for which data are publicly available. Accordingly, we assume that erosion risk is determined by a WHS's distance from the coast, the coastal material, mean wave height and sediment supply.

We use the coastline of the MCD[108] to calculate the shortest distance of each WHS from the coast. In several instances the coastline of the MCD considerably deviates from the actual coastline as detected with the help of Google Earth™, for example, around the cities of Trogir and Šibenik in Croatia or the city of Catania in

**Table 1 Scale values used for the components of the flood risk index and the erosion risk index**

| INDICATOR \ INDEX | 0 NOT AT RISK | 1 VERY LOW | 2 LOW | 3 MODERATE | 4 HIGH | 5 VERY HIGH |
|---|---|---|---|---|---|---|
| **FLOOD RISK** | | | | | | |
| Flood area [%] | 0 | > 0 | | | | ≥ 50 |
| Flood depth [m] | 0 | > 0 | | | | ≥ 1 |
| **EROSION RISK** | | | | | | |
| Distance [m] | > 500 | 500 | | | | < 10 |
| Coastal material | | rocky | - | muddy; rocky with pocket beaches | - | sandy |
| Mean wave height [m] | | 0.1 | | | | > 0.8 |
| Sediment supply [mg l⁻¹] | | 11.5 | | | | < 0.5 |

Italy. In these instances, we use the distance from the coastline of the global self-consistent, hierarchical, shoreline database version 2.3.7[120] (see dataset). We calculate the change in coastline due to SLR with the help of the SRTM data under the assumption that all areas below the amount of SLR in hydrological connection to the sea are inundated[121]. Again we interpolate linearly between elevation increments[60] and calculate the decrease in a WHS's distance from the coastline for each scenario and 10-year time step. Further, we use the MCD to assign the coastal material and mean wave height to each WHS based on the coastal segments attributed to the site. If more than one coastal material type or wave height is attributed to a WHS, we adopt the dominant one. To account for sediment supply, we use a newly created dataset of mean monthly total suspended matter (TSM) concentration. TSM is a measure of water turbidity in coastal locations that can be used as an indicator for sediment supply[122]. The original data were produced in the context of the GlobColour project and were calculated based on satellite imagery[123]. We spatially join the grid point data of the TSM to the coastal segments of the MCD closest to each grid point. If more than one grid point can be attributed to a segment, we calculate the mean of the points that extend along that segment. Subsequently, we attribute TSM values to each WHS, following the same procedure. We must point out that TSM represents sediment supply only to a limited degree as it does not include river bedload supplied at river mouths, which plays an important role in countering coastal erosion in the Mediterranean[124,125]. A dataset of bedload sediment transport is currently not available for the entire Mediterranean region. For the erosion risk index, we scale the four indicators linearly to values ranging from 0 (not at risk) to a maximum value of 5 (very high risk) based on scale values used in the literature that we adapt to the environmental conditions in the Mediterranean basin (Table 1). Accordingly, we assume a WHS to be at risk from coastal erosion if it is located at least within 500 m from the coast with the highest risk at or below 10 m distance[95], accounting for a twofold increase in observed erosion rates in the Mediterranean due to SLR[5,126]. For coastal material we use the scale values of refs.[5,96] and for mean wave height we adapt the values of ref[96]. For sediment supply we assume risk to be very high when the TSM concentration is below 0.5 mg l$^{-1}$. We calculate one erosion risk index (ERI) for each WHS based on Eq. (1), where $D$ stands for distance under the respective scenario and time step, $M$ for coastal material, mWH for mean wave height and TSM for total suspended matter. We follow the weighting used in Reeder-Myers[34], which is largely based on previous assessments[5,92,118] and we adjust it to the indicators included in this analysis, ensuring that the relative importance of each indicator remains unchanged. As sediment supply primarily plays a role in calm waters (i.e. beaches, wetlands, inlets) where it can get deposited[119], we exclude TSM from the risk index at WHS in rocky locations. In a last step, we scale the erosion risk index to a possible maximum value of 10:

$$\text{ERI}_{\text{rocky}} = (3D + 2M + \text{mWH}) \times \tfrac{1}{3}$$

$$\text{if } D > 500, \text{ERI} = 0 . \qquad (1)$$

$$\text{ERI}_{\text{other}} = (3D + 2M + \text{mWH} + \text{TSM}) \times \tfrac{1}{4}$$

**Code availability**. Spatial data processing was conducted in the Geographic Information System (GIS) software ArcGIS. The results of the spatial analysis were further processed in the software environment R to calculate the flood risk and erosion risk indices. The computer code of these calculations is available upon request.

## Data availability
The WHS datasets produced for this study are available in text format (CSV) and polygon vector format at https://doi.org/10.6084/m9.figshare.5759538 (ref.[32]).

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

## Acknowledgements

We thank Jorid Höffken for her help in assembling the UNESCO World Heritage data used in this study. S.B. was funded by a joint United Kingdom Natural Environment Research Council and United Kingdom Government Department of Business Energy and Industrial Strategy grant "ADJUST1.5," numbered NE/P01495X/1 and Natural Environment Research Council funded Innovation Fellowship NE/R00689X/1. We acknowledge financial support by the federal state of Schleswig-Holstein, Germany, within the funding programme Open Access Publikationsfonds.

## Author contributions

L.R. designed the research in close collaboration with A.T.V. and with support from S.B., J.H. and R.S.J.T. L.R. conducted the analysis and analysed the results in collaboration with A.T.V. L.R. wrote the manuscript with contributions from A.T.V., S.B. and J.H. All authors reviewed and edited the manuscript.

## Additional information

**Competing interests:** The authors declare no competing interests.

