## [Peer Review File · Nature Communications]

Reviewer #1 (Remarks to the Author):

In this manuscript, the authors seek to assess, at first order, the hazard sea-level rise, coastal flooding and erosion pose to UN World Heritage Sites in the Mediterranean Basin. To do this, they first produce a disaggregated georeferenced database of World Heritage Sites located in the Mediterranean Basin. They combine this database with a database of surge heights, three scenarios of sea-level rise for 2100, and a bathtub inundation model to index the exposure of each site to the 100-year flood when combined with sea-level rise. They also apply an index based on distance from the coast, coastal material, mean wave height, and sediment supply to index the erosion hazard.

The question the authors ask is an important and useful one, and the georeferenced database will no doubt prove useful to future researchers. The utility of the indices in their current form is less clear, but with some refinement, they could be made of clearer value.

I would suggest that the paper could be appropriate for publication with the following major changes:

1. A clearer explanation of the theoretical basis for the weightings used in the indices;
2. A justification for the linear combination of erosion risk and flood risk, or else dropping the linear combination and presenting these separately;
3. A clear explanation of how the authors' expect a planner to make use of the indices presented.

I myself am unclear of what #3 would be in the current presentation; however, it would be clear to me if the authors were to focus upon the temporal evolution of the flood risk index and/or its change under different emissions scenario, which -- irrespective of the relative value of different sites -- would allow the timing of adaptive measures to be prioritized and/or the value of global climate mitigation for each individual site to be assessed.

Utility of indices

The value of the authors' indices is less clear: they are comparable between sites, but since planning does not happen at a basin-wide scale, and in any case the authors tell us that the value of World Heritage Sites cannot be compared to one another, it is unclear what use they have in mind for this cross-site comparative ranking.

Moreover, the rationale for linearly combining the flood risk index and the erosion risk index is unclear, as is the rationale for the particular weightings used in the erosion risk index.

Conceivably, since each site can at least be compared to itself at a different point in time, the evolution of the flood risk index over time could highlight when the hazard at different sites becomes of concern, but the authors' use of a single end-of-century time point means it cannot happen based on current results.

The comparison of different scenarios might also help indicate the value of global climate mitigation for WHS protection, but since the scenarios used are the low end of the AR5 likely range for RCP 2.6, the high end of the AR5 likely range for RCP 8.5, and a third, higher scenario, the analysis in its present form does not lend itself to this comparison.

Indeed, from the current analysis it is unclear how much of the flood risk is due to sea-level rise, and how much exists under current conditions (i.e., 0.0 m AMSL).

Sea-level scenarios

I also find the authors' use of global sea-level rise (plus a completely unexplained GIA correction) in a highly localized context baffling. While it is certainly true that there is considerable uncertainty in the CMIP3 projections of dynamic sea level change in the Mediterranean (per ref. 90), the uncertainty in dynamic sea-level projections in the Mediterranean in CMIP5 does not appear to be unquantifiably large (17-83rd range width of about 30-45 cm, per Kopp et al 2014), and it's unclear why this uncertainty cannot simply be taken into account in the scenario definition. Other sources of uncertainty also are present, and yet are taken into account; and neglecting non-dynamic drivers of differences between global-mean sea-level change and regional sea level change (e.g., static equilibrium effects) seems unjustified. Per Kopp et al., 2014, the median ratio for RCP 8.5 in 2100 of climatically-driven (i.e., excluding GIA and tectonics) RSL rise to GMSL rise in the Mediterranean is about 0.8-0.9.

Indeed, the authors use as one of their sources the sea-level rise projections Jackson and Jevrejeva (2016). The main point of this paper is the production of regional sea-level projections. If these projections are not to the authors' liking, others (e.g., Kopp et al., 2014, 2017) also provide regionalized, probabilistic projections of RSL change around the world.

As an alternative approach, the authors might instead look at the flood index at different arbitrary but evenly spaced SLR levels (e.g., 0 to 2 m in 0.25 m spacing). This separates the question of flood risk from the question of sea-level rise timing and emissions sensitivity, which can then be addressed in a separable analysis. For example, the authors could for each site identify the RSL rise at which the site crosses a given flood risk threshold, and then draw upon the Jackson & Jevrejeva or Kopp et al. projections close to that site to determine the range of when that threshold might be crossed under different scenarios. This last identification could inform planners about when it would become necessary to prioritize coastal flood adaptation at different sites.

Tides

The authors should define 'extreme sea level' when it is first introduced. Throughout the manuscript, it is unclear whether extreme sea level is the sum of mean sea level and storm surge, or mean sea level, storm surge, and tide; if the former, it also means that the flood hazard is underestimated throughout.

SRTM

Although its use is perhaps unavoidable (unless there is a coastal LIDAR data set available for the Mediterranean), SRTM is a significantly flawed data set. Although the authors correctly describe its 'resolution' (the mission paper calls this 'vertical quantification') as 1 m, its vertical error is several meters (see the mission paper). Kulp and Strauss (2016, 2018) examine the implications of SRTM's limitations for flood risk assessment.

Kulp, S., & Strauss, B. H. (2016). Global DEM errors underpredict coastal vulnerability to sea level rise and flooding. *Frontiers in Earth Science*, 4, 36.

Kulp, S. A., & Strauss, B. H. (2018). CoastalDEM: A global coastal digital elevation model improved from SRTM using a neural network. *Remote Sensing of Environment*, 206, 231-239.

Minor comments

Line 63: Define the Low Elevation Coastal Zone when first mentioned. Do not bury the definition in Methods.

Line 73-75: Provide a brief explanation of the factors that cause ESL to be highest in these regions.

Line 76: With respect to tidal datum is the mean sea-level in AMSL defined?

Lines 76-88: It would be helpful to have these numbers visualized in a figure.

Line 91-97: Please provide a clearer physical explanation of the variability in the index.

Line 257-258: Explain how the surge heights are calculated.

Line 265-266: Explain the GIA correction.

Line 288: Although SLR is mentioned here, it does not appear to be included in the calculation of the erosion risk.

Figure 2: Please put figures comparable to these for the individual components of the indices in the SI.

Reviewer #2 (Remarks to the Author):

This paper presents an important contribution to recent studies that focus on the impacts of climate change on archaeology and cultural heritage around the world. The authors look at UNESCO World Heritage Sites around the Mediterranean, and examine the risk of flood and erosion based on different sea level rise scenarios. They use publically available data, modified as necessary to more

accurately reflect real world conditions, to determine that around three quarters of World Heritage Sites in this region are threatened by sea level rise, even under the most conservative estimates. They present their data and methodology in a clear, reproducible manner.

This type of analysis has become common in recent years, and the authors place their study in this larger context of research in Europe and around the world. Instead of focusing on all archaeological sites within a small region, which is a common approach, this study focuses on some of the highest profile cultural heritage sites in the world. In many ways, this is important. By highlighting the damage that will certainly occur at these iconic places, the authors draw attention to the dire condition of all cultural heritage in the coastal zone over the coming centuries. Moreover, such sites are more likely to receive the attention (i.e. funding) they need to mitigate those risks. That said, it is important to note that the vast majority of cultural heritage occurs in far less spectacular locations, and that the bulk of archaeological sites will be simply lost, along with the information and connections to the past that are contained in them. An evaluation of that loss clearly goes beyond the scope of the current paper, but the authors could point out that both the more humble and the more spectacular sites are important parts of human history.

This paper has more to say about the problem than about the solution—a criticism that could be equally leveled at almost every similar study of the past decade, including my own. However, I think the authors could say more in the discussion about others around the world who have tried to confront this problem, and offer more concrete solutions. In particular, they might look for examples in the work of Scottish Coastal Heritage at Risk program, and similar efforts in Ireland and England. The United States' National Park Service Cultural Resources Climate Change Strategy might also be useful. I am sure there are other worthy efforts around the world, but these are the ones I am most familiar with.

In sum, the authors have presented a concise, effective study of the impacts of sea level rise on World Heritage Sites around the Mediterranean. The methodology is thorough, but simple enough to be reproduced in many other parts of the world. I hope the next decade will produce similar research focused not only on evaluating the magnitude of the problem faced by cultural heritage, but also in documenting efforts to mitigate those problems.

--Leslie Reeder-Myers

Reviewer #3 (Remarks to the Author):

The paper proposed by the authors deals with an interesting aspect, the impact of coastal hazards on cultural heritage, which is of interest to the physical science community as well the archaeological community.

The idea of doing estimations at Mediterranean level is remarkable but the paper fails to convince the reader that the analyses done are sound and robust. Although at a qualitative level proposes some interesting hypotheses, at a quantitative level it is full of assumptions that shade a light of inaccuracy on the work.

FLOOD RISK

The paper in its current form refers to another paper that was under review:

Wolff, C. et al. A Mediterranean coastal database for assessing the impacts of sea-level rise and 391 associated hazards (under review).

Luckily I discovered that the paper has just been published as

<https://www.nature.com/articles/sdata201844#f3>

I went to consult that paper and I can take that for flood risk it produced an innovative dataset of return period levels for surges.

The main problem for this type of work is the fact that it relies heavily on the SRTM dataset which has the consensus view that it has a minimum vertical accuracy of 16 m absolute error at 90% confidence (Root Mean Square Error (RMSE) of 9.73 m) world-wide.

The paper draws some strong conclusion for sites located at deltaic areas like the Po and the Rhone Delta, which have an elevation of few metres above MSL when are not below MSL. I have the feeling that with this type of inaccuracy of the dataset, the inundation model has not the capability of distinguish real flooding from artefacts. Moreover, the morphological data on the shoreline does not include anything about the level of protection, neither the real elevation of dykes and embankments, nor their characteristics.

Indeed the authors admit that :” We do not account for existing flood protection measures in our analysis due to a lack of consistent region-wide data.” However, for some regions like the Po area and its southern coastline these information could have been obtained from local knowledge data

and at least used to carry out a sensitivity analysis on the results. This coastline is heavily protected and it seems unlikely that the current level of defence would not be adapted to rising sea-level.

EROSION RISK

If the flood risk, after proper evidence of reliability in data-rich regions, may seem acceptable, the erosion risk makes little sense. The fact that risk is determined by a site's distance from the coast is a simplistic viewpoint.

A strong role in this evaluation is played by the "sediment supply" parameter, making reference to

Vafeidis, A.T. et al. The DIVA database documentation. Available at <http://ftp.demris.nl/outgoing/dinascoast/Docs/Papers/Vafeidis,%20A.T.%20et%20al,%202005%202542%20DIVA%20dbase%20documentation.pdf> (2005).

This is a broken link and in any case, in all the DIVA model's output I have never seen meaningful information on sediment supply.

Moreover the statement "The sediment supply index includes qualitative information on sediment supply; a high index (maximum of 5) represents low sediment supply and a low index (minimum of 1) represents high sediment supply" confirms my worries. It is a totally subjective index.

The paper as it is not acceptable and scientifically sound. I propose to the authors to re-write the paper considering:

- 1) To eliminate the Erosion Index
- 2) To produce evidence that the Flooding Index is reliable by taking a data-rich example and use a Lidar dataset to compare the output of an SRTM approach with a detailed Lidar approach and carry out a sensitivity analysis for the flood index.

Manuscript “Mediterranean UNESCO World Heritage at risk from coastal flooding and erosion due to sea-level rise”

We would like to thank all reviewers and the editor for their valuable and constructive comments. We have carefully considered all comments and have substantially revised the manuscript in response to the points that the reviewers have raised. Major changes include:

- repeating the model runs, using regional SLR projections, including 10-year time steps to account for the temporal evolution of the risk indices and for changes in the coastline due to permanent inundation induced by SLR*
- providing further literature-based justification for the use of the erosion risk index and the weighting of its individual parameters*
- an extended discussion on the relevance of our results for policymakers on regional scales*
- a concise discussion on the limitations of this study (DEM, coastal defences)*

Below you can find the responses to each point raised by the reviewers (replies in red italics).

Reviewers' comments:

Reviewer #1

I would suggest that the paper could be appropriate for publication with the following major changes:

1. A clearer explanation of the theoretical basis for the weightings used in the indices
2. A justification for the linear combination of erosion risk and flood risk, or else dropping the linear combination and presenting these separately
3. A clear explanation of how the authors' expect a planner to make use of the indices presented.

I myself am unclear of what #3 would be in the current presentation; however, it would be clear to me if the authors were to focus upon the temporal evolution of the flood risk index and/or its change under different emissions scenario, which -- irrespective of the relative value of different sites -- would allow the timing of adaptive measures to be prioritized and/or the value of global climate mitigation for each individual site to be assessed.

We would like to thank Reviewer #1 for the constructive and clear comments. We have conducted additional model runs and further analysis based on the methodological changes made to address the comments. We believe that the manuscript has improved considerably. Below we respond to each comment and describe the steps we have undertaken to address them.

Utility of indices

- 1.) The value of the authors' indices is less clear: they are comparable between sites, but since planning does not happen at a basin-wide scale, and in any case the authors tell us that the value of World Heritage Sites cannot be compared to one another, it is unclear what use they have in mind for this cross-site comparative ranking.

We agree with the reviewer in that planning primarily takes place at national scale, which makes intra-national comparison of WHS at risk particularly important. Especially in countries with a large number of WHS at risk from either of the two hazards national-level assessments can provide insights into how to prioritise adaptation, where to allocate resources and where to carry out further research based on more complex modelling approaches and local-scale data. However, planning also takes place on a Mediterranean-wide scale as established in 1976 in the Barcelona Convention for the Protection of the Marine Environment and the Coastal Region of the Mediterranean. The Barcelona Convention was updated in 1995 and provides the basis of the Mediterranean Action Plan that is part of the UNEP Regional Seas Programme. One goal of the Barcelona Convention is to promote integrated management of the Mediterranean coastal zone. Considerable efforts have been undertaken in recent years with the aim to facilitate basin-wide planning and management such as the Protocol on Integrated Coastal Zone Management in the Mediterranean (2008)¹, the Mediterranean Strategy for Sustainable Development 2016-2025 (2016)² and the Regional Climate Change Adaptation Framework for the Mediterranean Marine and Coastal Areas (2017)³. Some of these policy documents explicitly mention the need for managing cultural heritage in the region²⁻⁶. To address these points we have extended the methods (lines 296-300) and discussion (lines 136-142) sections where we clarify the usefulness of the risk indices for policymakers at national and basin scales.

- 2.) Moreover, the rationale for linearly combining the flood risk index and the erosion risk index is unclear, as is the rationale for the particular weightings used in the erosion risk index.

We agree with the comment of the reviewer that the combined index was not meaningful in supporting decision-makers. We have therefore removed the linear combination of the two indices. Regarding the weightings of the erosion risk index: these are based on the current literature that assesses coastal erosion⁶⁻¹² and cultural heritage at risk from coastal erosion¹³⁻¹⁵ on regional scales and specifically on the weightings used in Reeder-Myers et al. (2015)¹³. As we use exclusively publicly available data, we have adjusted the weighting of the erosion risk indicators to the number of indicators used in this analysis, ensuring that the relative importance of each indicator remains unchanged. We have added text to clarify and support the weighting assumptions (lines 409-412).

- 3.) Conceivably, since each site can at least be compared to itself at a different point in time, the evolution of the flood risk index over time could highlight when the hazard at different sites becomes of concern, but the authors' use of a single end-of-century time point means it cannot happen based on current results.

The comparison of different scenarios might also help indicate the value of global climate mitigation for WHS protection, but since the scenarios used are the low end of the AR5 likely range for RCP 2.6, the high end of the AR5 likely range for RCP 8.5, and a third, higher scenario, the analysis in its present form does not lend itself to this comparison.

Indeed, from the current analysis it is unclear how much of the flood risk is due to sea-level rise, and how much exists under current conditions (i.e., 0.0 m AMSL).

Thank you for this comment. We have now included the temporal evolution of both risk indices. More specifically, we have calculated the flood risk index for the base year 2000 that represents current conditions (0.0 m AMSL) until the year 2100, in 10-year time steps (we have done the same for the erosion risk index; see comments 14 and 20). To do so, we have revised the method used to calculate the floodplain in order to overcome the limitation of the 1 m vertical resolution of the SRTM DEM. We now use the full elevation increments of the extreme sea levels (ESL) assigned to each WHS and linearly interpolate between respective elevation increments to guarantee an adequate estimation of the floodplain¹⁶. Please see lines 350-359 in the methods for further clarification.

In order to help indicate the value of global mitigation, we calculate the floodplain for three RCPs (RCP2.6, RCP4.5, RCP8.5) instead of two, using the median (50th percentile) projections of the regionalised sea-level rise (SLR) projections of Kopp et al. (2017)¹⁷ (see our response to comment 4). For the high-end (HE) scenario, we additionally use the 95th percentile of RCP8.5. We have changed the respective section in the methods (lines 343-346), we describe current conditions as well as temporal changes depending on the scenario in the results (lines 78-99; Figure 2, Figure 4) and we discuss the value of the temporal evolution of risk for adaptation (lines 144-149, Figure 6) and mitigation (lines 158-166).

Sea-level scenarios

- 4.) I also find the authors' use of global sea-level rise (plus a completely unexplained GIA correction) in a highly localized context baffling. While it is certainly true that there is considerable uncertainty in the CMIP3 projections of dynamic sea level change in the Mediterranean (per ref. 90), the uncertainty in dynamic sea-level projections in the Mediterranean in CMIP5 does not appear to be unquantifiably large (17-83rd range width of about 30-45 cm, per Kopp et al 2014), and it's unclear why this uncertainty cannot simply be taken into account in the scenario definition. Other sources of uncertainty also are present, and yet are taken into account; and neglecting non-dynamic drivers of differences between global-mean sea-level change and regional sea level change (e.g., static equilibrium effects)

seems unjustified. Per Kopp et al., 2014, the median ratio for RCP 8.5 in 2100 of climatically-driven (i.e., excluding GIA and tectonics) RSL rise to GMSL rise in the Mediterranean is about 0.8-0.9.

Indeed, the authors use as one of their sources the sea-level rise projections Jackson and Jevrejeva (2016). The main point of this paper is the production of regional sea-level projections. If these projections are not to the authors' liking, others (e.g., Kopp et al., 2014, 2017) also provide regionalized, probabilistic projections of RSL change around the world.

As an alternative approach, the authors might instead look at the flood index at different arbitrary but evenly spaced SLR levels (e.g., 0 to 2 m in 0.25 m spacing). This separates the question of flood risk from the question of sea-level rise timing and emissions sensitivity, which can then be addressed in a separable analysis. For example, the authors could for each site identify the RSL rise at which the site crosses a given flood risk threshold, and then draw upon the Jackson & Jevrejeva or Kopp et al. projections close to that site to determine the range of when that threshold might be crossed under different scenarios. This last identification could inform planners about when it would become necessary to prioritize coastal flood adaptation at different sites.

We have repeated our analysis using the regionalized SLR projections by Kopp et al. 2017 (K17), which are based on the projections produced in Kopp et al. 2014 (K14). These projections account for Glacial Isostatic Adjustment based on the ICE-5G VM2-90 model by Peltier et al. 2004¹⁸. We selected the K17 projections as they are available for tide gauge locations (like in K14), which are primarily located along the northern coast of the Mediterranean, as well as for grid points with a resolution of 2° by 2° that are spread across the entire Mediterranean basin. Further, the K17 projections are available in 10-year time steps from 2010 to 2300 from which we have selected the years of interest to our analysis. We spatially joined the K17 data to the coastal segments of the Mediterranean Coastal Database (MCD) based on the point closest to each segment to be able to calculate the ESL for each SLR scenario (RCP2.6, RCP4.5, RCP8.5, HE) and time step (2010-2100). We have changed the manuscript accordingly (lines 339-348).

Tides

- 5.) The authors should define 'extreme sea level' when it is first introduced. Throughout the manuscript, it is unclear whether extreme sea level is the sum of mean sea level and storm surge, or mean sea level, storm surge, and tide; if the former, it also means that the flood hazard is underestimated throughout.

The storm surge heights that we use are based on the Global Tide and Surge Reanalysis (GTSR) dataset by Muis et al. 2016 that accounts for storm surges and tides¹⁹. Hence we define 'extreme sea level' as the sum of mean sea level and storm surge height. We have added a definition to the

manuscript (lines 78-80) and changed lines 329-332 in the methods for clarification.

SRTM

- 6.) Although its use is perhaps unavoidable (unless there is a coastal LIDAR data set available for the Mediterranean), SRTM is a significantly flawed data set. Although the authors correctly describe its 'resolution' (the mission paper calls this 'vertical quantification') as 1 m, its vertical error is several meters (see the mission paper). Kulp and Strauss (2016, 2018) examine the implications of SRTM's limitations for flood risk assessment.

Kulp, S., & Strauss, B. H. (2016). Global DEM errors underpredict coastal vulnerability to sea level rise and flooding. *Frontiers in Earth Science*, 4, 36.

Kulp, S. A., & Strauss, B. H. (2018). CoastalDEM: A global coastal digital elevation model improved from SRTM using a neural network. *Remote Sensing of Environment*, 206, 231-239.

We are aware of the limitations of the SRTM data but, to our knowledge, there is currently no other region-wide DEM available with better characteristics. At the same time the SRTM DEM has been extensively used in large-scale flood modelling^{16,19-23}. Indeed, the DEM has an absolute vertical error of 16 m. Importantly, the relative error is considerably smaller (<10 m), while the absolute vertical error has been shown to be smaller in coastal areas^{24,25}. In their paper, Kulp & Strauss²⁶ also conclude that the SRTM data are currently the most reliable publicly available elevation data.

LiDAR data are in few cases available for certain parts of the Mediterranean and using those for a limited number of locations would compromise the consistency of our results across the basin. Further, processing such data for the entire Mediterranean basin is extremely time and resource consuming and goes beyond the scope of this analysis. As we consider publicly available data as an important element of this study for adopting the approach to other regions, we are unable to use the newly created CoastalDEM²⁷ as it is not freely available. Although the CoastalDEM could provide a more accurate assessment of risk, so far no studies have been published that test these data and allow comparisons with other results. We have modified and extended the discussion pointing out the limitations of the SRTM data (lines 248-261), also citing Kulp & Strauss (2016) for further discussion of the SRTM DEM and Kulp & Strauss (2018) as a potential alternative to the use of the SRTM data.

Minor comments

- 7.) Line 63: Define the Low Elevation Coastal Zone when first mentioned. Do not bury the definition in Methods.

We have added the definition of the LECZ when it is first mentioned (lines 67-68).

- 8.) Line 73-75: Provide a brief explanation of the factors that cause ESL to be highest in these regions.

As the focus of this study lies on the results of the risk indices rather than the spatial patterns of the ESL, we have removed the respective paragraph from the results section. We provide a short explanation of the factors causing ESL to be highest in lines 96-98.

- 9.) Line 76: With respect to tidal datum is the mean sea-level in AMSL defined?

We define the mean sea level based on the vertical datum of the SRTM data that are referenced to the EGM96 geoid. As the storm surge data are referenced to the mean sea level, we have converted them to EGM96 with the help of the mean dynamic ocean topography (MDT) which provides the difference between mean sea level and the geoid. We use the MDT of the MCD based on Rio et al. 2014²⁸. Recent studies have made similar corrections^{22,26,27,29}. We have added text in lines 333-337 for clarification.

- 10.) Lines 76-88: It would be helpful to have these numbers visualized in a figure.

We have added Figure 2 to visualise the temporal evolution of the flood risk index and have produced a similar figure for the erosion risk index (Figure 4). Further we present the changes in risk between 2000 and 2100 (under the HE scenario) for each WHS in Figure 3 and Figure 5. Please note that we have restructured the results section with the aim to emphasise the difference between current and potential future conditions.

- 11.) Line 91-97: Please provide a clearer physical explanation of the variability in the index.

We provide an explanation of the factors that lead to the variability of the erosion risk index in lines 118-121 and we have added a reference to two figures in SI1-4 and SI1-5 that visualise the erosion risk components distance (years 2000 and 2100 under the HE scenario), coastal material, mean wave height, and sediment supply (see also comment 15).

- 12.) Line 257-258: Explain how the surge heights are calculated.

The MCD provides a downscaled version of the Global Tide and Surge Reanalysis (GTSR) dataset produced by Muis et al. 2016. A detailed explanation of how the surge heights were calculated can

be found in Muis et al. 2016¹⁹. We have added text and a reference to Muis et al. 2016 to clarify this point (lines 329-333).

13.)Line 265-266: Explain the GIA correction.

As the regionalized SLR projections of Kopp et al. 2017¹⁷ account for GIA based on the ICE-5G VM2-90 model by Peltier et al. 2004¹⁸ (referred to in line 342), we have removed the GIA correction from the calculation of the flood risk index.

14.)Line 288: Although SLR is mentioned here, it does not appear to be included in the calculation of the erosion risk.

We account for an increase in erosion risk with SLR by calculating the decreasing distance of a WHS from the coast for each SLR scenario and 10-year time step (2010-2100). We use a similar method to the one used for calculating the floodplain (comment 3): we calculate the change in coastline due to SLR with the help of the SRTM data under the assumption that all areas below the amount of SLR are inundated³⁰ and interpolate linearly between elevation increments. We have changed the methods accordingly (lines 386-390).

Please note that we have changed the maximum distance for a WHS to be at risk from erosion to 500 m (lines 402-405 and Table 1) as we did not account for a decrease in WHS distance from the coast due to SLR in the previous version of the manuscript.

Please see our response to comment 20 of Reviewer #3 for further details regarding the erosion risk index.

15.)Figure 2: Please put figures comparable to these for the individual components of the indices in the SI.

We have added one figure of the flood risk components flood area and flood depth for the years 2000 and 2100 under the HE scenario to SI1-3. We have added an additional figure of the erosion risk components distance (years 2000 and 2100 under the HE scenario), coastal material, mean wave height, and sediment supply to SI1-4 and SI1-5. The results of other scenarios and time steps are available in text format (CSV) in SI2 (flood risk), SI3 (erosion risk).

Reviewer #2

This paper presents an important contribution to recent studies that focus on the impacts of climate change on archaeology and cultural heritage around the world. The authors look at UNESCO World Heritage Sites around the Mediterranean, and examine the risk of flood and erosion based on different sea level rise scenarios. They use publically available data, modified as necessary to more

accurately reflect real world conditions, to determine that around three quarters of World Heritage Sites in this region are threatened by sea level rise, even under the most conservative estimates. They present their data and methodology in a clear, reproducible manner.

16.) This type of analysis has become common in recent years, and the authors place their study in this larger context of research in Europe and around the world. Instead of focusing on all archaeological sites within a small region, which is a common approach, this study focuses on some of the highest profile cultural heritage sites in the world. In many ways, this is important. By highlighting the damage that will certainly occur at these iconic places, the authors draw attention to the dire condition of all cultural heritage in the coastal zone over the coming centuries. Moreover, such sites are more likely to receive the attention (i.e. funding) they need to mitigate those risks. That said, it is important to note that the vast majority of cultural heritage occurs in far less spectacular locations, and that the bulk of archaeological sites will be simply lost, along with the information and connections to the past that are contained in them. An evaluation of that loss clearly goes beyond the scope of the current paper, but the authors could point out that both the more humble and the more spectacular sites are important parts of human history.

We thank the reviewer for raising this point. In this study we concentrate on UNESCO World Heritage as, different from cultural resources not inscribed in the World Heritage List, basic data and documentation are available from the UNESCO website. Of course only a small number of our common heritage is inscribed in the UNESCO World Heritage List, representing the most spectacular examples of heritage located around the world. We have added text to the discussion to emphasise that these sites are more likely to be protected from the risks of coastal flooding and erosion and to point out that other cultural heritage not inscribed in the list receive far less attention and will possibly be lost in the future (lines 275-281).

17.) This paper has more to say about the problem than about the solution—a criticism that could be equally leveled at almost every similar study of the past decade, including my own. However, I think the authors could say more in the discussion about others around the world who have tried to confront this problem, and offer more concrete solutions. In particular, they might look for examples in the work of Scottish Coastal Heritage at Risk program, and similar efforts in Ireland and England. The United States' National Park Service Cultural Resources Climate Change Strategy might also be useful. I am sure there are other worthy efforts around the world, but these are the ones I am most familiar with.

We agree with the reviewer – this is indeed a common problem with this type of studies. At the same time however we believe that it is important to highlight and better frame the problem of WHS at risk from coastal hazards as, in many cases, this has not been done before. Our study aims to initiate discussion rather than finding solutions, while at the same time providing new results and insights into the topic.

In response to this comment we have reviewed the examples stated above and have integrated them into the discussion, acknowledging recent efforts in managing cultural heritage (lines 185-191). Although we are not able to make concrete suggestions of specific adaptation measures for WHS, we additionally discuss the examples of adaptation at coastal heritage that we could find in the current literature, along with some ideas regarding potential adaptation options as well as limits to adaptation, which are particularly important with regard to WHS (lines 170-208).

In sum, the authors have presented a concise, effective study of the impacts of sea level rise on World Heritage Sites around the Mediterranean. The methodology is thorough, but simple enough to be reproduced in many other parts of the world. I hope the next decade will produce similar research focused not only on evaluating the magnitude of the problem faced by cultural heritage, but also in documenting efforts to mitigate those problems.

--Leslie Reeder-Myers

We would like to thank the reviewer for her positive and constructive comments and we are hopeful that we have responded to her comments in a satisfactory manner.

Reviewer #3

The paper proposed by the authors deals with an interesting aspect, the impact of coastal hazards on cultural heritage, which is of interest to the physical science community as well the archaeological community.

The idea of doing estimations at Mediterranean level is remarkable but the paper fails to convince the reader that the analyses done are sound and robust. Although at a qualitative level proposes some interesting hypotheses, at a quantitative level it is full of assumptions that shade a light of inaccuracy on the work.

We have carefully considered the reviewer's suggestions and have conducted additional analysis to address the points raised. We have also substantially rewritten large parts of the manuscript to provide additional information regarding the issues of concern.

FLOOD RISK

18.)The paper in its current form refers to another paper that was under review:

Wolff, C. et al. A Mediterranean coastal database for assessing the impacts of sea-level rise and associated hazards (under review).

Luckily I discovered that the paper has just been published as <https://www.nature.com/articles/sdata201844#f3>

I went to consult that paper and I can take that for flood risk it produced an innovative dataset of return period levels for surges.

The main problem for this type of work is the fact that it relies heavily on the SRTM dataset which has the consensus view that it has a minimum vertical accuracy of 16 m absolute error at 90% confidence (Root Mean Square Error (RMSE) of 9.73 m) world-wide.

The paper draws some strong conclusion for sites located at deltaic areas like the Po and the Rhone Delta, which have an elevation of few metres above MSL when are not below MSL. I have the feeling that with this type of inaccuracy of the dataset, the inundation model has not the capability of distinguish real flooding from artefacts.

The reviewer is correct in pointing out the limitations of the SRTM, which is also a point raised by Reviewer #1 (comment 6). We are aware of the limitations of the SRTM data but, to our knowledge, there is currently no other region-wide DEM available with better characteristics. At the same time the SRTM DEM has been extensively used in large-scale flood modelling^{16,19–23}. Indeed, the DEM has an absolute vertical error of 16 m. Importantly, the relative error is considerably smaller (<10 m), while the absolute vertical error has been shown to be smaller in coastal areas^{24,25}. A recent study assessing the limitations of the SRTM data for coastal vulnerability assessments²⁶ also concludes that the SRTM data are currently the most reliable publicly available elevation data.

LiDAR data are in few cases available for certain parts of the Mediterranean and using those for a limited number of locations would compromise the consistency of our results across the basin. Further, processing such data for the entire Mediterranean basin is extremely time and resource consuming and goes beyond the scope of this analysis. As we consider publicly available data as an important element of this study for adopting the approach to other regions, we are unable to use the newly created CoastalDEM²⁷ as it is not freely available. Although the CoastalDEM could provide a more accurate assessment of risk, so far no studies have been published that test these data and allow comparisons with other results. We have modified and extended the discussion pointing out the limitations of the SRTM data (lines 248-261), also citing Kulp & Strauss (2016) for further discussion of the SRTM DEM and Kulp & Strauss (2018) as a potential alternative to the use of the SRTM data.

Please note that we have revised the method used to calculate the floodplain in order to overcome the limitation of the 1 m vertical resolution of the SRTM DEM. We now use the full elevation increments of the ESL assigned to each WHS and linearly interpolate between respective elevation

increments to guarantee an adequate estimation of the floodplain¹⁶ (lines 353-359). We have used the same approach for the erosion risk calculation (lines 386-390 and response to comment 20).

19.) Moreover, the morphological data on the shoreline does not include anything about the level of protection, neither the real elevation of dykes and embankments, nor their characteristics.

Indeed the authors admit that :” We do not account for existing flood protection measures in our analysis due to a lack of consistent region-wide data.” However, for some regions like the Po area and its southern coastline these information could have been obtained from local knowledge data and at least used to carry out a sensitivity analysis on the results. This coastline is heavily protected and it seems unlikely that the current level of defence would not be adapted to rising sea-level.

Accounting for coastal defences is a very complex task that goes beyond the scope of this analysis. It is noteworthy that most large-scale impact assessment studies do not account for protection, with very few exceptions such as the DIVA model which models adaptation responses¹⁶. Further, as data availability on coastal protection standards at global and regional scales is extremely limited, accounting for coastal defences for specific locations/regions would compromise the consistency of our results across the basin. The only region-wide data available are those assembled by Scussolini et al. (2016) that describe protection standards on an administrative unit basis³¹. These data are of limited use to our study as they include data on protection standards for parts of Croatia, Slovenia and Italy only.

Due to the scale of the study and the methods used a sensitivity analysis would not provide much additional information as parameterisation of protection would simply result in uniform reductions in risk. This sensitivity is also indirectly included in our indices through the parameters of flood area and flood depth. Nevertheless, we believe that the results of this study provide a good first estimate of WHS at risk as there is only limited protection for very few locations along the Mediterranean coast and even less so for WHS (with Venice and its Lagoon as a notable exception where protection structures are being established; see lines 237-240). We have added text to the methods (lines 360-363) and emphasise that our results may lead to an overestimation¹⁶ of risk in locations with existing coastal protection measures (lines 230-243).

EROSION RISK

20.) If the flood risk, after proper evidence of reliability in data-rich regions, may seem acceptable, the erosion risk makes little sense. The fact that risk is determined by a site’s distance from the coast is a simplistic viewpoint.

A strong role in this evaluation is played by the “sediment supply” parameter, making reference to Vafeidis, A.T. et al. The DIVA database documentation. Available at

<http://ftp.demis.nl/outgoing/dinascoast/Docs/Papers/Vafeidis,%20A.T.%20et%20al,%202005%2542%20DIVA%20dbase%20documentation.pdf> (2005).

This is a broken link and in any case, in all the DIVA model's output I have never seen meaningful information on sediment supply.

Moreover the statement "The sediment supply index includes qualitative information on sediment supply; a high index (maximum of 5) represents low sediment supply and a low index (minimum of 1) represents high sediment supply" confirms my worries. It is a totally subjective index.

We have carefully considered the reviewer's comment. As we do see value in the index (and since Reviewers #1 & #2 have not deemed it unnecessary) we have revised it and have addressed the points that the reviewer has raised. Specifically we have made the following changes:

- a) We account for an increase in erosion risk with SLR by calculating the decreasing distance of a WHS from the coast for each SLR scenario and 10-year time step (2010-2100). We use a similar method to the one used for calculating the floodplain (comments 3 & 18): we calculate the change in coastline due to SLR with the help of the SRTM data under the assumption that all areas below the amount of SLR are inundated³⁰ and interpolate linearly between elevation increments (lines 386-390). We have also changed the maximum distance for a WHS to be at risk from erosion to 500 m (lines 402-405 and Table 1) as we did not account for a decrease in WHS distance from the coast due to SLR in the previous version of the manuscript.*
- b) As sediment supply plays an important role for coastal erosion in the Mediterranean^{5,32,33}, we use a newly created dataset of mean monthly Total Suspended Matter (TSM) concentration to represent local sediment availability³⁴. TSM is a measure of the turbidity of the water in coastal locations and has been produced based on satellite imagery as part of the GlobColour project³⁵. We spatially join the TSM data to the MCD and subsequently attribute TSM to each WHS. As sediment supply mainly plays a role in calm waters where it can be deposited⁵, we exclude TSM from the erosion risk index at WHS in rocky locations. Please see lines 393-399, 406-407 and 412-414 for further details.*
- c) Instead of assigning discrete classes (0-5) to each erosion risk indicator, we have scaled the indicators to a continuous scale with a possible maximum value of 5 (lines 400-402). This way we ensure that as little information as possible is lost during the calculation of the erosion risk index. For this purpose, we have used scale values from the literature that we adapted to the environmental conditions in the Mediterranean basin (e.g. the mean wave height in the Mediterranean is much lower than those values proposed in Reeder et al. (2012)¹⁴ or in Mavromatidi et al. (2018)⁷). Previous studies have also adjusted the scale values based on*

their case study region, see e.g. Refs.^{7,13,14}. Please see comment 2 and lines 409-412 for the rationale of the weightings used in the index.

To our knowledge, erosion risk has not been analysed with regard to UNESCO World Heritage in previous work and our results can therefore provide a first estimate of those WHS potentially at risk from erosion due to SLR. We are aware that one main limitation of index-based approaches is their rather qualitative nature and the fact that they rely on a certain degree of subjectivity^{11,36-39}. Nevertheless, index-based approaches assessing the risk of coastal erosion are well-established in current research^{6-12,40}, with some studies explicitly analysing cultural heritage at risk from coastal erosion¹³⁻¹⁵. The risk index allows us to assess WHS without attaching monetary value to them. This approach is particularly suitable for first-order assessments on regional scale to support adaptation planning^{10,11,40}. We do not pursue the aim to encourage policymakers to base their decisions regarding specific adaptation measures on this index, but rather to use it as an indication of erosion risk potential. Especially in countries with a large number of WHS at risk, these results can provide insights into how to prioritise adaptation, where to allocate resources and where to carry out further research based on more complex modelling approaches and local-scale data. To clarify the points stated above, we discuss the usefulness of both indices in lines 136-149, 296-300.

The paper as it is not acceptable and scientifically sound. I propose to the authors to re-write the paper considering:

- 1) To eliminate the Erosion Index
- 2) To produce evidence that the Flooding Index is reliable by taking a data-rich example and use a Lidar dataset to compare the output of an SRTM approach with a detailed Lidar approach and carry out a sensitivity analysis for the flood index.

References

1. UNEP/MAP. *Protocol on integrated coastal zone management in the Mediterranean. Protocole relatif à la gestion intégrée des zones côtières de la Méditerranée = Protocolo relativo a la gestion integrada de las zonas costeras del Mediterraneo* (United Nation Environment Programme, [s.l.], 2008).
2. UNEP/MAP. *Mediterranean strategy for sustainable development 2016-2025. Investing in environmental sustainability to achieve social and economic development* (Plan Bleu, Regional Activity Centre, Valbonne, 2016).
3. UNEP/MAP. *Regional Climate Change Adaptation Framework for the Mediterranean Marine and Coastal Areas* (UNEP/MAP, Athens, Greece, 2017).
4. Benoit, G. & Comeau, A. *A sustainable future for the Mediterranean. The Blue Plan's environment and development outlook* (Earthscan, London, 2005).

5. UNEP/MAP. State of the Mediterranean marine and coastal environment. Available at https://wedocs.unep.org/bitstream/handle/20.500.11822/364/sommcer_eng.pdf?sequence=4&isAllowed=y (2012).
6. Satta, A., Venturini, S., Puddu, M., Firth, J. & Lafitte, A. Strengthening the Knowledge Base on Regional Climate Variability and Change: Application of a Multi-Scale Coastal Risk Index at Regional and Local Scale in the Mediterranean. Available at http://planbleu.org/sites/default/files/publications/multi-scale_coastal_risk_index.pdf (2015).
7. Mavromatidi, A., Briche, E. & Claeys, C. Mapping and analyzing socio-environmental vulnerability to coastal hazards induced by climate change. An application to coastal Mediterranean cities in France. *Cities* **72**, 189–200 (2018).
8. Gornitz, V.M., Daniels, R.C., White, T.W. & Birdwell, K.R. The Development of a Coastal Risk Assessment Database: Vulnerability to Sea-Level Rise in the U.S. Southeast. *Journal of Coastal Research*, 327–338 (1994).
9. Boruff, B.J., Emrich, C. & Cutter, S.L. Erosion Hazard Vulnerability of US Coastal Counties. *Journal of Coastal Research* **215**, 932–942 (2005).
10. Torresan, S., Critto, A., Rizzi, J. & Marcomini, A. Assessment of coastal vulnerability to climate change hazards at the regional scale. The case study of the North Adriatic Sea. *Nat. Hazards Earth Syst. Sci.* **12**, 2347–2368 (2012).
11. Mclaughlin, S. & Cooper, J.A.G. A multi-scale coastal vulnerability index. A tool for coastal managers? *Environmental Hazards* **9**, 233–248 (2010).
12. Pendleton, E.A., Thieler, R.E. & Williams, J.S. Coastal Vulnerability Assessment of Cape Hatteras National Seashore (CAHA) to Sea-Level Rise. Available at <https://pubs.usgs.gov/of/2004/1064/images/pdf/caha.pdf> (2004).
13. Reeder-Myers, L.A. Cultural Heritage at Risk in the Twenty-First Century. A Vulnerability Assessment of Coastal Archaeological Sites in the United States. *The Journal of Island and Coastal Archaeology* **10**, 436–445 (2015).
14. Reeder, L.A., Rick, T.C. & Erlandson, J.M. Our disappearing past. A GIS analysis of the vulnerability of coastal archaeological resources in California's Santa Barbara Channel region. *J Coast Conserv* **16**, 187–197 (2012).
15. Daire, M.-Y. *et al.* Coastal Changes and Cultural Heritage (1). Assessment of the Vulnerability of the Coastal Heritage in Western France. *The Journal of Island and Coastal Archaeology* **7**, 168–182 (2012).
16. Hinkel, J. *et al.* Coastal flood damage and adaptation costs under 21st century sea-level rise. *Proceedings of the National Academy of Sciences of the United States of America* **111**, 3292–3297 (2014).
17. Kopp, R.E. *et al.* Evolving Understanding of Antarctic Ice-Sheet Physics and Ambiguity in Probabilistic Sea-Level Projections. *Earth's Future* **5**, 1217–1233 (2017).
18. Peltier, W.R. Global glacial isostasy and the surface of the ice-age earth. The ICE-5G (VM2) Model and GRACE. *Annu. Rev. Earth Planet. Sci.* **32**, 111–149 (2004).
19. Muis, S., Verlaan, M., Winsemius, H.C., Aerts, J.C.J.H. & Ward, P.J. A global reanalysis of storm surges and extreme sea levels. *Nature communications* **7**, 11969 (2016).
20. Jongman, B., Ward, P.J. & Aerts, J.C. Global exposure to river and coastal flooding. Long term trends and changes. *Global Environmental Change* **22**, 823–835 (2012).
21. Muis, S., Guneralp, B., Jongman, B., Aerts, J.C.J.H. & Ward, P.J. Flood risk and adaptation strategies under climate change and urban expansion: A probabilistic analysis using global data. *The Science of the total environment* **538**, 445–457 (2015).
22. Muis, S. *et al.* A comparison of two global datasets of extreme sea levels and resulting flood exposure. *Earth's Future* **5**, 379–392 (2017).

23. Wolff, C., Vafeidis, A.T., Lincke, D., Marasmi, C. & Hinkel, J. Effects of Scale and Input Data on Assessing the Future Impacts of Coastal Flooding. An Application of DIVA for the Emilia-Romagna Coast. *Front. Mar. Sci.* **3**, 34 (2016).
24. Luana, S., Hou, X. & Wang, Y. Assessing the Accuracy of SRTM Dem and Aster Gdem Datasets for the Coastal Zone of Shandong Province, Eastern China. *Polish Maritime Research* **22**, 15–20 (2015).
25. Gorokhovich, Y. & Voustianiouk, A. Accuracy assessment of the processed SRTM-based elevation data by CGIAR using field data from USA and Thailand and its relation to the terrain characteristics. *Remote Sensing of Environment* **104**, 409–415 (2006).
26. Kulp, S. & Strauss, B.H. Global DEM Errors Underpredict Coastal Vulnerability to Sea Level Rise and Flooding. *Front. Earth Sci.* **4**, 4823 (2016).
27. Kulp, S.A. & Strauss, B.H. CoastalDEM. A global coastal digital elevation model improved from SRTM using a neural network. *Remote Sensing of Environment* **206**, 231–239 (2018).
28. Rio, M.-H., Mulet, S. & Picot, N. Beyond GOCE for the ocean circulation estimate. Synergetic use of altimetry, gravimetry, and in situ data provides new insight into geostrophic and Ekman currents. *Geophys. Res. Lett.* **41**, 8918–8925 (2014).
29. Ramirez, J.A., Lichter, M., Coulthard, T.J. & Skinner, C. Hyper-resolution mapping of regional storm surge and tide flooding. Comparison of static and dynamic models. *Nat Hazards* **82**, 571–590 (2016).
30. Antonioli, F. *et al.* Sea-level rise and potential drowning of the Italian coastal plains. Flooding risk scenarios for 2100. *Quaternary Science Reviews* **158**, 29–43 (2017).
31. Scussolini, P. *et al.* FLOPROS. An evolving global database of flood protection standards. *Nat. Hazards Earth Syst. Sci.* **16**, 1049–1061 (2016).
32. Ninfo, A., Ciavola, P. & Billi, P. The Po Delta is restarting progradation: geomorphological evolution based on a 47-years Earth Observation dataset. *Scientific reports* **8**, 3457 (2018).
33. Besset, M., Anthony, E.J. & Sabatier, F. River delta shoreline reworking and erosion in the Mediterranean and Black Seas. The potential roles of fluvial sediment starvation and other factors. *Elem Sci Anth* **5**, 54 (2017).
34. Schuerch, M. *et al.* Future response of global coastal wetlands to sea level rise. *Nature* (accepted for publication).
35. Doerffer, R. & Schiller, H. The MERIS Case 2 water algorithm. *International Journal of Remote Sensing* **28**, 517–535 (2010).
36. Moel, H. de, van Alphen, J. & Aerts, J. C. J. H. Flood maps in Europe – methods, availability and use. *Nat. Hazards Earth Syst. Sci.* **9**, 289–301 (2009).
37. Preston, B.L., Yuen, E.J. & Westaway, R.M. Putting vulnerability to climate change on the map. A review of approaches, benefits, and risks. *Sustain Sci* **6**, 177–202 (2011).
38. Reckien, D. What is in an index? Construction method, data metric, and weighting scheme determine the outcome of composite social vulnerability indices in New York City. *Reg Environ Change* **18**, 1439–1451 (2018).
39. Balica, S.F., Wright, N.G. & van der Meulen, F. A flood vulnerability index for coastal cities and its use in assessing climate change impacts. *Nat Hazards* **64**, 73–105 (2012).
40. Daly, C. A Framework for Assessing the Vulnerability of Archaeological Sites to Climate Change. Theory, Development, and Application. *Conservation and Management of Archaeological Sites* **16**, 268–282 (2014).

Reviewer #3 (Remarks to the Author):

Dear authors,

I find the manuscript much better than before.

Regarding your statement in the rebuttal letter which I copy below:

"As sediment supply plays an important role for coastal erosion in the Mediterranean^{5,32,33}, we use a newly created dataset of mean monthly Total Suspended Matter (TSM) concentration to represent local sediment availability³⁴. TSM is a measure of the turbidity of the water in coastal locations and has been produced based on satellite imagery as part of the GlobColour project³⁵. We spatially join the TSM data to the MCD and subsequently attribute TSM to each WHS. As sediment supply mainly plays a role in calm waters where it can be deposited⁵, we exclude TSM from the erosion risk index at WHS in rocky locations. Please see lines 393-399, 406-407 and 412-414 for further details."

I am still dubious about using as proxy for sediment supply a value of suspended sediment (e.g. turbidity), obtained from satellite measurements. What really counts on counteracting coastal erosion is river bedload at the mouths (generally sand), while what you obtain from satellite is generally fine suspended sediment, which is dispersed offshore.

Maybe it would be wise to make a statement that you are aware of the limits of this dataset of turbidity data. What really lacks is a global dataset of bedload sediment transport for rivers.

Indeed the literature sources you cite in the rebuttal evidenced the importance of sediment supply in Mediterranean deltaic areas. However, I have not seen in the new version of the document the sources you cite:

32. Ninfo, A., Ciavola, P. & Billi, P. The Po Delta is restarting progradation: geomorphological evolution based on a 47-years Earth Observation dataset. Scientific reports 8, 3457 (2018).

33. Besset, M., Anthony, E.J. & Sabatier, F. River delta shoreline reworking and erosion in the Mediterranean and Black Seas. The potential roles of fluvial sediment starvation and other factors. *Elem Sci Anth* 5, 54 (2017).

Maybe the reference list was not updated, I suggested to do so. In any case I appreciate your awareness of the problem of sediment supply.

Reviewer #4 (Remarks to the Author):

I have been asked to examine the authors' revisions and responses to Review #1's comments and concerns regarding the use of global sea level rise scenarios in a local context.

Reviewer #1 suggested that it was inappropriate to use global sea level projections with some sort of local (presumably GIA) correction. I have not seen the original manuscript, but I would agree. The revised manuscript has addressed this concern by re-doing their analysis with the Kopp et al. (2017) probabilistic projections which are provided at tide gauge sites and on a 2 deg by 2 deg grid. I believe that the modifications that the authors have made sufficiently address Reviewer #1's concerns on this matter.

In addition to focusing on this specific aspect of the manuscript, I have a few additional minor comments below.

It is worth noting though that work has shown that future 100-yr flood heights are biased low when you consider that sea level is rising at an uncertain rate (Buchanan et al., 2016, *Climatic Change*). While the Mediterranean was not explicitly discussed in Buchanan et al (2016), it would be worthwhile for this paper to be cited as it will be important for site-specific adaptation efforts.

Line specific comments:

Line 81: "HE" has not yet been defined.

Line 113: I think you are missing some words in this sentence. I don't follow it.

Figure 4b: this is the change in erosion risk index? Please make this more explicit.

Manuscript “Mediterranean UNESCO World Heritage at risk from coastal flooding and erosion due to sea-level rise”

We would like to thank the reviewers for their remaining comments. We have addressed them in the manuscript using the track changes tool and we have responded to each comment below (red italics). Please be aware that we have also implemented the editor’s comments in the manuscript.

Reviewer #3

Dear authors,

I find the manuscript much better than before.

1. Regarding your statement in the rebuttal letter which I copy below:
"As sediment supply plays an important role for coastal erosion in the Mediterranean^{5,32,33}, we use a newly created dataset of mean monthly Total Suspended Matter (TSM) concentration to represent local sediment availability³⁴. TSM is a measure of the turbidity of the water in coastal locations and has been produced based on satellite imagery as part of the GlobColour project³⁵. We spatially join the TSM data to the MCD and subsequently attribute TSM to each WHS. As sediment supply mainly plays a role in calm waters where it can be deposited⁵, we exclude TSM from the erosion risk index at WHS in rocky locations. Please see lines 393-399, 406-407 and 412-414 for further details."

I am still dubious about using as proxy for sediment supply a value of suspended sediment (e.g. turbidity), obtained from satellite measurements. What really counts on counteracting coastal erosion is river bedload at the mouths (generally sand), while what you obtain from satellite is generally fine suspended sediment, which is dispersed offshore.

Maybe it would be wise to make a statement that you are aware of the limits of this dataset of turbidity data. What really lacks is a global dataset of bedload sediment transport for rivers.

Indeed the literature sources you cite in the rebuttal evidenced the importance of sediment supply in Mediterranean deltaic areas. However, I have not seen in the new version of the document the sources you cite:

32. Ninfo, A., Ciavola, P. & Billi, P. The Po Delta is restarting progradation: geomorphological evolution based on a 47-years Earth Observation dataset. *Scientific reports* 8, 3457 (2018).

33. Besset, M., Anthony, E.J. & Sabatier, F. River delta shoreline reworking and erosion in the Mediterranean and Black Seas. The potential roles of fluvial sediment starvation and other factors. *Elem Sci Anth* 5, 54 (2017).

Maybe the reference list was not updated, I suggested to do so. In any case I appreciate your awareness of the problem of sediment supply.

We would like to thank the reviewer for raising this point. We have now added text to the manuscript to emphasise the limits of the TSM data and the importance of sediment supply in Mediterranean deltaic locations (lines 426-429) along with references to Ninfo et al. 2018 and Besset et al. 2017 (Refs. 123 and 124 in the manuscript).

Reviewer #4

I have been asked to examine the authors' revisions and responses to Review #1's comments and concerns regarding the use of global sea level rise scenarios in a local context.

Reviewer #1 suggested that it was inappropriate to use global sea level projections with some sort of local (presumably GIA) correction. I have not seen the original manuscript, but I would agree. The revised manuscript has addressed this concern by re-doing their analysis with the Kopp et al. (2017) probabilistic projections which are provided at tide gauge sites and on a 2 deg by 2 deg grid. I believe that the modifications that the authors have made sufficiently address Reviewer #1's concerns on this matter.

We would like to thank Reviewer #4 for his/her constructive comments. Below we respond to each point raised.

In addition to focusing on this specific aspect of the manuscript, I have a few additional minor comments below.

2. It is worth noting though that work has shown that future 100-yr flood heights are biased low when you consider that sea level is rising at an uncertain rate (Buchanan et al., 2016, Climatic Change). While the Mediterranean was not explicitly discussed in Buchanan et al (2016), it would be worthwhile for this paper to be cited as it will be important for site-specific adaptation efforts.

After consulting Buchanan et al. 2016, we have added text to the discussion, pointing out the need to account for a potential low bias in future return flood heights in local-scale assessments due to uncertainties regarding the rate of sea-level rise (lines 233-236). We have also added a citation to the paper (Ref. 58 in the manuscript).

Line specific comments:

3. Line 81: "HE" has not yet been defined.

We have added text to define "HE" (line 103).

4. Line 113: I think you are missing some words in this sentence. I don't follow it.

We have rephrased the sentence for clarification (lines 136-141).

5. Figure 4b: this is the change in erosion risk index? Please make this more explicit.

We are unsure about this point as we believe that we explicitly describe the contents of Figure 4b. Please see the title of Figure 4b, the text of the y-axis and the caption accompanying Figure 4 for clarification. However, in response to the previous comment (comment 4) we have changed the manuscript section that describes Figure 4b (lines 136-141); we hope that this is clearer now.